# Adaptation of spontaneous activity in the developing visual cortex

**Marina E Wosniack[1,2], Jan H Kirchner[1,2], Ling-Ya Chao[1], Nawal Zabouri[3], Christian Lohmann[3,4], Julijana Gjorgjieva[1,2]\***

[1]Computation in Neural Circuits Group, Max Planck Institute for Brain Research, Frankfurt, Germany; [2]School of Life Sciences Weihenstephan, Technical University of Munich, Freising, Germany; [3]Netherlands Institute for Neuroscience, Amsterdam, Netherlands; [4]Center for Neurogenomics and Cognitive Research, Vrije Universiteit, Amsterdam, Netherlands

**Abstract** Spontaneous activity drives the establishment of appropriate connectivity in different circuits during brain development. In the mouse primary visual cortex, two distinct patterns of spontaneous activity occur before vision onset: local low-synchronicity events originating in the retina and global high-synchronicity events originating in the cortex. We sought to determine the contribution of these activity patterns to jointly organize network connectivity through different activity-dependent plasticity rules. We postulated that local events shape cortical input selectivity and topography, while global events homeostatically regulate connection strength. However, to generate robust selectivity, we found that global events should adapt their amplitude to the history of preceding cortical activation. We confirmed this prediction by analyzing *in vivo* spontaneous cortical activity. The predicted adaptation leads to the sparsification of spontaneous activity on a slower timescale during development, demonstrating the remarkable capacity of the developing sensory cortex to acquire sensitivity to visual inputs after eye-opening.

**\*For correspondence:**
gjorgjieva@brain.mpg.de

**Competing interests:** The authors declare that no competing interests exist.

## Introduction

The impressive ability of the newborn brain to respond to its environment and generate coordinated output without any prior experience suggests that brain networks undergo substantial organization, tuning and coordination even as animals are still in the womb, driven by powerful developmental mechanisms. These broadly belong to two categories: activity-independent mechanisms, involving molecular guidance cues and chemoaffinity gradients which establish the initial coarse connectivity patterns at early developmental stages (*Feldheim and O'Leary, 2010*; *Goodhill, 2016*), and activity-dependent plasticity mechanisms which continue with refinement of this initially imprecise connectivity into functional circuits that can execute diverse behaviors in adulthood (*Ackman and Crair, 2014*; *Richter and Gjorgjieva, 2017*; *Thompson et al., 2017*). Non-random patterns of spontaneous activity drive these refinements and act as training inputs to the immature circuits before the onset of sensory experience. Many neural circuits in the developing brain generate spontaneous activity, including the retina, hippocampus, cortex, and spinal cord (reviewed in *Blankenship and Feller, 2010*; *Wang and Bergles, 2015*). This activity regulates a plethora of developmental processes such as neuronal migration, ion channel maturation, and the establishment of precise connectivity (*Huberman et al., 2008*; *Moody and Bosma, 2005*; *Kirkby et al., 2013*; *Godfrey and Swindale, 2014*), and perturbing this activity impairs different aspects of functional organization and axonal refinement (*Cang et al., 2005a*; *Xu et al., 2011*; *Burbridge et al., 2014*). These studies firmly demonstrate that spontaneous activity is necessary and instructive for the emergence of specific and distinct patterns of neuronal connectivity in the developing nervous system.

Recent *in vivo* recordings in the developing sensory cortex have found that the spatiotemporal properties of spontaneous activity, including frequency, synchronicity, amplitude and spatial spread, depend on the studied region and developmental age (*Golshani et al., 2009*; *Rochefort et al., 2009*; *Gribizis et al., 2019*). These studies have shown that the generation and propagation of spontaneous activity in the intact cortex depend on input from different brain areas. For instance, activity from the sensory periphery substantially contributes to the observed activity patterns in the developing cortex, but there are other independent sources of activity within the cortex itself (*Ackman et al., 2012*; *Siegel et al., 2012*; *Hanganu et al., 2006*; *Gribizis et al., 2019*). Two-photon imaging of spontaneous activity in the *in vivo* mouse primary visual cortex before eye-opening (post-natal days, P8-10) has demonstrated that there are two independently occurring patterns of spontaneous activity with different sources and spatiotemporal characteristics. Peripheral events driven by retinal waves (*Feller et al., 1996*; *Blankenship and Feller, 2010*) spread in the cortex as low-synchronicity local events (L-events), engaging a relatively small number of the recorded neurons. In contrast, events intrinsic to the cortex that are unaffected by manipulation of retinal waves spread as highly synchronous global events (H-events), activating a large proportion of the recorded neurons (*Siegel et al., 2012*).

We know relatively little about the information content of these local and global patterns of spontaneous cortical activity relevant for shaping local and brain-wide neural circuits. Specifically, it is unknown whether spontaneous activity from different sources affects distinct aspects of circuit organization, each providing an independent instructive signal, or if L- and H-events cooperate to synergistically guide circuit organization. Therefore, using experimentally characterized properties of spontaneous activity in the visual cortex *in vivo* at P8-10, we developed a biologically plausible, yet analytically tractable, theoretical framework to determine the implications of this activity on normal circuit development with a focus on the topographic refinement of connectivity and the emergence of stable receptive fields.

We postulated that peripheral L-events play a key role in topographically organizing receptive fields in the cortex, while H-events regulate connection strength homeostatically, operating in parallel to network refinements by L-events. We considered that H-events are ideally suited for this purpose because they maximally activate many neurons simultaneously, and hence lack topographic information that can be used for synaptic refinement. We studied two prominent activity-dependent plasticity rules to investigate the postulated homeostatic function of H-events, the Hebbian covariance rule (*Miller et al., 1989*; *Miller, 1994*; *Lee et al., 2002*; *Sejnowski, 1977*) and the Bienenstock-Cooper-Munro (BCM) rule (*Bienenstock et al., 1982*). In the Hebbian covariance rule, simultaneous pre- and postsynaptic activation (e.g. during L-events) triggers the selective potentiation of synaptic connections, while postsynaptic activation without presynaptic input (e.g. during H-events) leads to the unselective depression of all connections. In the BCM rule, H-events dynamically regulate potentiation and depression. However, both rules generate receptive fields that have either refinement or topography defects. Therefore, we proposed that H-events might be self-regulating, with amplitudes that adapt to the levels of recent cortical activity. Indeed, we found evidence of this adaptation in spontaneous activity recorded in the developing visual cortex (*Siegel et al., 2012*). Besides generating topographically refined receptive fields, this adaptation leads to the sparsification of cortical spontaneous activity over a prolonged timescale of development as in the visual and somatosensory cortex (*Rochefort et al., 2009*; *Golshani et al., 2009*). Therefore, our work proposes that global, cortically generated activity in the form of H-events rapidly adapts to ongoing network activity, supporting topographic organization of connectivity and maintaining synaptic strengths in an operating regime.

## Results

### A network model for connectivity refinements driven by spontaneous activity

How spontaneous activity instructs network refinements between the sensory periphery and the visual cortex depends on two aspects: the properties of spontaneous activity and the activity-dependent learning rules that translate these properties into specific changes in connectivity. We first characterized spontaneous activity in the mouse primary visual cortex before eye-opening, and

investigated two prominent learning rules to organize connectivity in a network model of the thalamus and visual cortex.

Spontaneous activity recorded *in vivo* using two-photon $Ca^{2+}$ imaging exhibits two independently occurring patterns: network events originating in the retina and propagating through the thalamus, and network events generated in the cortex (*Siegel et al., 2012*; *Figure 1A*). These two types of events were first identified by a cluster analysis based on event amplitude and jitter (a measure of synchrony; *Siegel et al., 2012*). The analysis identified a participation rate criterion to separate network events into local low-synchronicity (L-) events generated in the retina, where 20–80% of the neurons in the field of view are simultaneously active, and global high-synchronicity (H-) events intrinsic to the cortex, where nearly all (80–100%) cortical neurons are simultaneously active. This same 80% participation rate criterion was recently validated both at the single-cell and population levels (*Leighton et al., 2020*). We first confirmed differences in specific features of the recorded spontaneous events (*Siegel et al., 2012*), and also characterized novel aspects (*Figure 1B*). In particular, L-events have a narrow distribution of amplitudes and inter-event intervals (IEI, the inverse of firing frequency) that follow an exponential-like distribution. H-events have a broader distribution of amplitudes with higher values and IEIs that follow a long-tailed distribution with higher values relative to L-events. We found that L- and H-events have similar durations.

Next, we built a model that incorporates these two different patterns of spontaneous activity to investigate the potentially different roles that L- and H-events might play in driving connectivity refinements between the thalamus and the visual cortex (*Figure 1C*). We used a one-dimensional feedforward network model – a microcircuit motivated by the small region of cortex imaged experimentally – composed of two layers, an input (presynaptic) layer corresponding to the sensory periphery (the thalamus) and a target (postsynaptic) layer corresponding to the primary visual cortex (*Figure 2A*). Cortical activity $v$ in the model is generated by two sources (*Figure 2B*; *Table 1*). First, L-events, $u$, activate a fraction between 20% and 80% of neighboring thalamic cells (also referred to as the L-event size) and drive the cortex through the weight matrix, $W$. Second, H-events, $v^{spon}$, activate the majority of the cortical cells (a fraction between 80% and 100%, also referred to as the

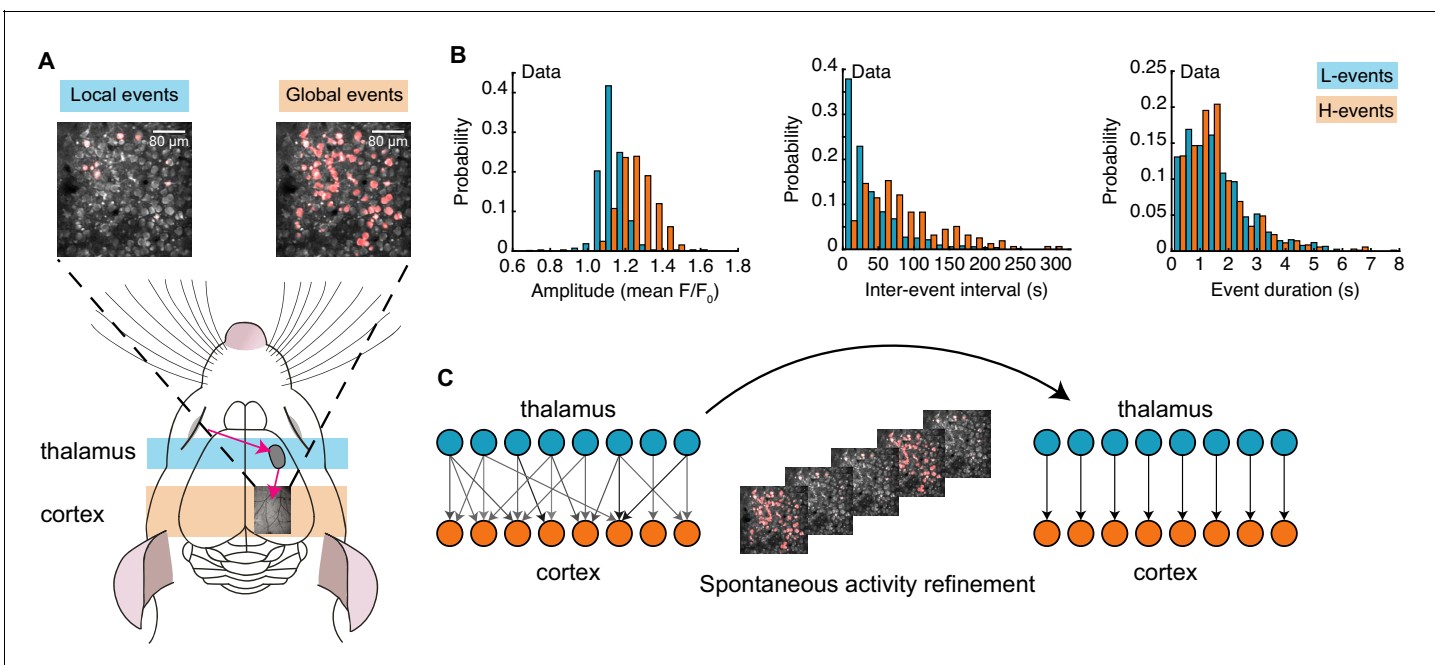

**Figure 1.** Spontaneous activity patterns in early postnatal development. (**A**) Two distinct patterns of spontaneous activity recorded *in vivo* in the visual cortex of young mice before eye-opening (P8-10). Blue shading denotes local low-synchronicity (L-) events generated by the retina; orange shading denotes global high-synchronicity (H-) events generated by the cortex. Activated neurons during each event are shown in red. (**B**) Distributions of different event properties (amplitude, inter-event interval, and event duration). Amplitude was measured as changes in fluorescence, relative to baseline, F/F0. (**C**) Network schematic: thalamocortical connections are refined by spontaneous activity. The initially broad receptive fields with weak synapses evolve into a stable configuration with strong synapses organized topographically.

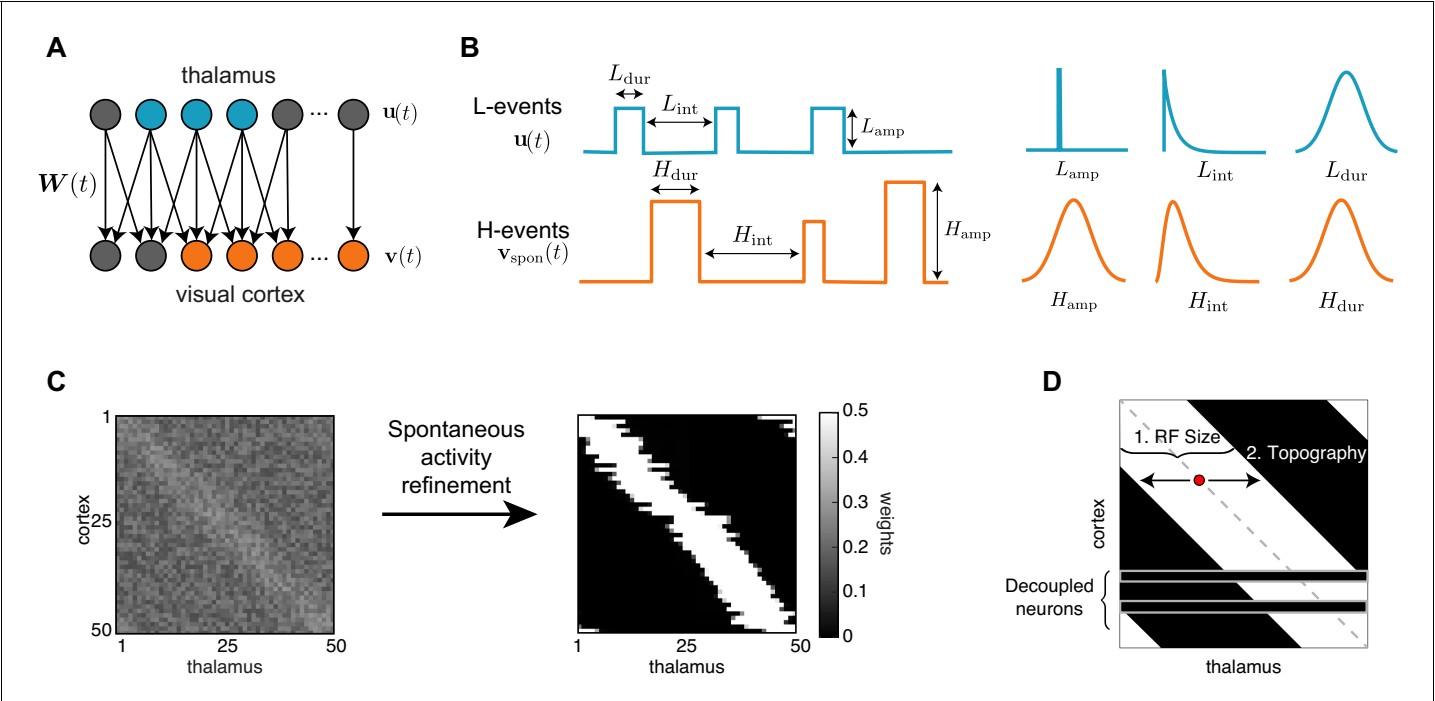

**Figure 2.** A network model of thalamocortical connectivity refinements. (**A**) A feedforward network with an input layer of thalamic neurons $\boldsymbol{u}(t)$ connected to an output layer of cortical neurons $\boldsymbol{v}(t)$ by synaptic weights $\boldsymbol{W}(t)$. (**B**) Properties of L- and H-events in the model (amplitude $L_{\mathrm{amp}}, H_{\mathrm{amp}}$, inter-event interval $L_{\mathrm{int}}, H_{\mathrm{int}}$ and duration $L_{\mathrm{dur}}, H_{\mathrm{dur}}$) follow probability distributions extracted from data (*Siegel et al., 2012*) (see *Table 1*). (**C**) Initially weak all-to-all connectivity with a small topographic bias along the diagonal (left) gets refined by the spontaneous activity events (right). (**D**) Evaluating network refinement through receptive field statistics (see Materials and methods). We quantify two properties: (1) the receptive field size and (2) the topography, which quantifies on average how far away the receptive field center of each cortical cell (red dot) is from the diagonal (dashed gray line).

H-event size). We used a rate-based unit with a membrane time constant $\tau_m$ and linear activation function consistent with the coarse temporal structure of spontaneous activity during development, carrying information on the order of hundreds of milliseconds (*Gjorgjieva et al., 2009*; *Butts and Kanold, 2010*; *Richter and Gjorgjieva, 2017*):

$$\tau_m \frac{\mathrm{d}\boldsymbol{v}(t)}{\mathrm{d}t} = -\boldsymbol{v}(t) + \boldsymbol{W}(t)\boldsymbol{u}(t) + \boldsymbol{v}^{\mathrm{spon}}(t). \tag{1}$$

To investigate the refinement of network connectivity during development, we studied the evolution of synaptic weights using plasticity rules operating over long timescales identified experimentally (*Butts et al., 2007*; *Winnubst et al., 2015*). First, we examined a classical Hebbian plasticity rule where coincident presynaptic thalamic activity and postsynaptic cortical activity in the form of L-events leads to synaptic potentiation. We postulated that H-events act homeostatically and maintain synaptic weights in an operating regime by depressing the majority of synaptic weights in the absence of peripheral drive. Because they activate most cortical neurons simultaneously, H-events lack the potential to drive topographical refinements. Their postulated homeostatic action resembles synaptic depression through downscaling, as observed in response to highly correlated network activity, for instance, upon blocking inhibition (*Turrigiano and Nelson, 2004*), or during slow-wave sleep (*Tononi and Cirelli, 2006*). Therefore, to the Hebbian rule we added a non-Hebbian term that depends only on the postsynaptic activity, with a proportionality constant that controls the relative amount of synaptic depression. This differs from other Hebbian covariance plasticity rules for the generation of weight selectivity, which include non-Hebbian terms that depend on both pre- and postsynaptic activity (*Lee et al., 2002*; *Mackay and Miller, 1990*) and is mathematically related to models of heterosynaptic plasticity (*Chistiakova et al., 2014*; *Lynch et al., 1977*; *Zenke et al., 2015*). Hence, the change in synaptic weight between cortical neuron *j* and thalamic neuron *i* is given by:

**Table 1.** List of parameters used in the model unless stated otherwise.

| Name | Value/Distribution | Description |
|---|---|---|
| **Network** | | |
| $N_u$ | 50 | Number of thalamic neurons |
| $N_v$ | 50 | Number of cortical neurons |
| $T$ | 50,000 | Simulation length [s] |
| **Weights** | | |
| $w_{ini}$ | U(0.15,0.25) | Range of initial weights (U: uniform dist.) |
| $s$ | 0.05 | Amplitude of Gaussian bias |
| $\sigma_s$ | 4 | Spread of Gaussian bias |
| $w_{max}$ | 0.5 | Weight saturation limit |
| **L-events** | | |
| $L_{amp}$ | 1.0 | Amplitude (equivalently, binary neuron) |
| $L_{pct}$ | U(20%,80%) | Percentage of thalamic cells activated |
| $L_{dur}$ | $\mathcal{N}(0.15, 0.015)$ | Mean duration [s] ($\mathcal{N}$: Gaussian dist.) |
| $L_{int}$ | Exp(1.5) | Mean inter-event interval [s] (Exp: exponential dist.) |
| **H-events** | | |
| $H_{amp}$ | $\mathcal{N}(6, 2)$ | Amplitude |
| $H_{pct}$ | U(80%,100%) | Percentage of cortical cells activated |
| $H_{dur}$ | $\mathcal{N}(0.15, 0.015)$ | Mean duration [s] |
| $H_{int}$ | Gamma(3.5, 1.0) | Mean inter-event interval [s] (Gamma: Gamma dist.) |
| **Time constants** | | |
| $\tau_m$ | 0.01 | Membrane time constant [s] |
| $\tau_w$ | 500 | Weight-change time constant for Hebbian covariance rule [s] |
| $\tau_w$ | 1000 | Weight-change time constant for BCM rule [s] |
| $\tau_\theta$ | 20 | Output threshold time constant for BCM rule [s] |
| $\tau_\eta$ | 1 | Adaptation time constant [s] |

$$\tau_w \frac{\mathrm{d}w_{ji}(t)}{\mathrm{d}t} = v_j(t)(u_i(t) - \theta_u), \qquad (2)$$

where $\tau_w$ is the learning time constant and $\theta_u$ the proportionality constant in the non-Hebbian term, which we refer to as the 'input threshold'. The activity time constant $\tau_m$ is much faster than the learning time constant, $\tau_m \ll \tau_w$, which allows us to separate timescales and to study how network activity on average affects learning (see Appendix). Interestingly, in this Hebbian covariance rule, the input threshold together with H-events effectively implement a subtractive constraint (see Appendix: 'Normalization constraints'). Subtractive normalization preserves the sum of all weights by subtracting from each weight a constant amount independent of each weight strength and is known to generate selectivity and refined receptive fields (*Miller and MacKay, 1994*). This is in contrast to the alternative multiplicative normalization, which generates graded and unrefined receptive fields where most correlated inputs are represented (*Miller and MacKay, 1994*) and hence was not considered here.

Additionally, we investigated the BCM learning rule, which can induce weight stability and competition without imposing constraints in the weights, and hence generate selectivity in postsynaptic neurons which experience patterned inputs (*Bienenstock et al., 1982*). For instance, the BCM framework can explain the emergence of ocular dominance (neurons in primary visual cortex being selective for input from one of the two eyes) and orientation selectivity in the visual system (*Cooper et al., 2004*). An important property of the BCM rule is its ability to homeostatically regulate the balance between potentiation and depression of all incoming inputs into a given neuron

depending on how far away the activity of that neuron is from some target level. The change in synaptic weight between cortical neuron $j$ and thalamic neuron $i$ is given by:

$$\tau_w \frac{\mathrm{d}w_{ji}(t)}{\mathrm{d}t} = v_j(t)u_i(t)(v_j(t) - \theta_v^j(t)), \tag{3}$$

where

$$\tau_\theta \frac{\mathrm{d}\theta_v^j(t)}{\mathrm{d}t} = -\theta_v^j(t) + \frac{v_j^2(t)}{v_0} \tag{4}$$

describes the threshold $\theta_v^j(t)$ between depression and potentiation which slides as a function of postsynaptic activity, $v_0$ is the target rate of the cortical neurons and $\tau_\theta$ the sliding threshold time constant. According to this rule, synaptic weight change is Hebbian in that it requires coincident pre- and postsynaptic activity, as is only the case during L-events. H-events induce no direct plasticity in the network because of the absence of presynaptic activation, but they still trigger synaptic depression indirectly by increasing the threshold between potentiation and depression.

Based on experimental measurements of the extent of thalamocortical connectivity at different developmental ages (*López-Bendito, 2018*), we assumed that initial network connectivity was weak and all-to-all, such that each cortical neuron was innervated by all thalamic neurons. To account for the activity-independent stage of development guided by molecular guidance cues and chemoaffinity gradients, a small bias was introduced to the initial weight matrix to generate a coarse topography in the network, where neighboring neurons in the thalamus project to neighboring neurons in the cortex and preserve spatial relationships (*Figure 2C*, left). Following connectivity refinements through spontaneous activity and plasticity, a desired outcome is that the network achieves a stable topographic configuration (*Figure 2C*, right) where each cortical neuron receives input only from a neighborhood of thalamic neurons.

To evaluate the success of this process, we quantified two properties. First, the receptive field size defined as the average number of thalamic neurons that strongly innervate a cortical cell (*Figure 2D*). We normalized the receptive field size to the total number of thalamic cells, so that it ranges from 0 (no receptive field, all cortical cells decouple from the thalamus) to 1 (each cortical cell receives input from all the thalamic cells, all weights potentiate leading to no selectivity). We also quantified the topography of the final receptive field (*Figure 2D* and Materials and methods), which evaluates how well the initial bias is preserved in the final network connectivity. The topography ranges from 0 (all cortical neurons connect to the same set of thalamic inputs) to 1 (perfect topography relative to the initial bias). We note that the lack of initial connectivity bias did not disrupt connectivity refinements and receptive field formation but could not on its own establish topography (*Figure 3—figure supplement 1A*).

## Spontaneous cortical H-events disrupt topographic connectivity refinement in the Hebbian covariance and BCM plasticity rules

Both the Hebbian and the BCM learning rules are known to generate selectivity with patterned input stimuli (*Mackay and Miller, 1990*; *Bienenstock et al., 1982*), and we confirmed that L-events on their own can refine receptive fields in both scenarios (*Figure 3—figure supplement 2*). We found that including H-events in the Hebbian covariance rule requires that the parameters of the learning rule and the properties of H-events (the input threshold $\theta_u$ and the inter-event interval $H_{\mathrm{int}}$) follow a tight relationship to generate selective and refined receptive fields (*Figure 3A,C*, left). For a narrow range of $H_{\mathrm{int}}$, weight selectivity emerges, but with some degree of decoupling between pre- and postsynaptic neurons (*Figure 3A*, middle). Outside of this narrow functional range, individual cortical neurons are either non-selective (*Figure 3A*, left) or decoupled from the thalamus (*Figure 3A*, right). These results are robust to changes in the participation rates of L- and H-events. For instance, when H-events involve 70–100% of cortical neurons, the percent of outcomes with selective receptive fields increases slightly to 19.8% (compared to 14.0% when H-events involve 80–100% of cortical neurons), while the percent of outcomes with decoupled cortical neurons increases to 60.4% (compared to 43.6% when H-events involve 80–100% of cortical neurons), reinforcing the idea that H-events are detrimental to receptive field refinements. In comparison, including H-events in the BCM learning rule does not decouple pre- and postsynaptic neurons (*Figure 3B*) and selectivity can

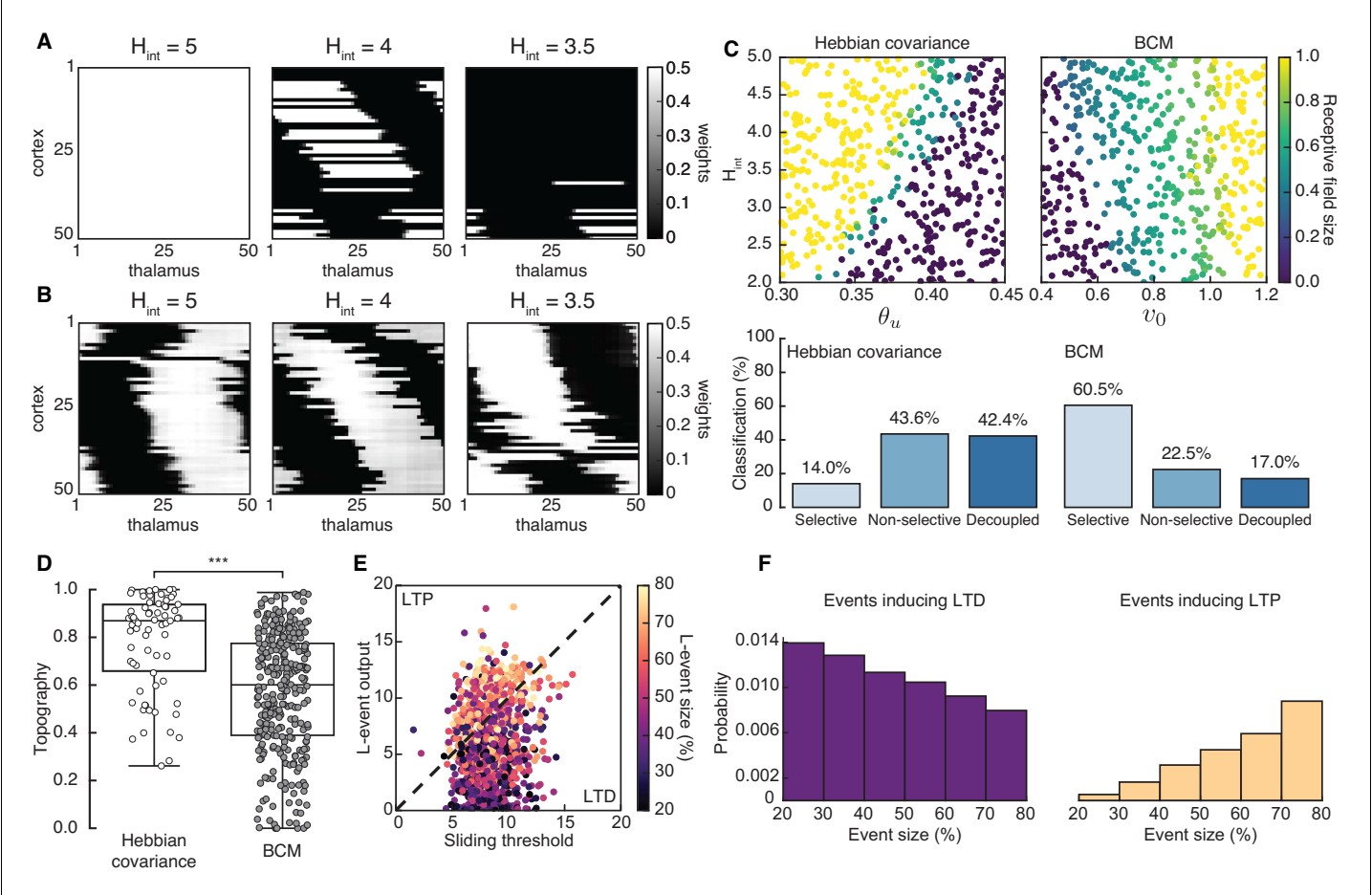

**Figure 3.** Spontaneous cortical events disrupt receptive field refinement. (**A**) Receptive fields generated by the Hebbian covariance rule with input threshold $\theta_u = 0.4$ and decreasing $H_{\mathrm{int}}$. (**B**) Receptive fields generated by the BCM rule with target rate $v_0 = 0.7$ and decreasing $H_{\mathrm{int}}$. (**C**) Top: Receptive field sizes obtained from 500 Monte Carlo simulations for combinations of $H_{\mathrm{int}}$ and $\theta_u$ for the Hebbian covariance rule (left) and $H_{\mathrm{int}}$ and $v_0$ for the BCM rule (right). Bottom: Percentage of simulation outcomes classified as 'selective' when the average receptive field size is smaller than one and larger than 0, 'non-selective' when the average receptive field size is equal to 1, and 'decoupled' when the average receptive field size is 0 for the two rules. (**D**) Topography of receptive fields classified as selective in C. Horizontal line indicates median, the box is drawn between the 25th and 75th percentile, whiskers extend above and below the box to the most extreme data points that are within a distance to the box equal to 1.5 times the interquartile range and points indicate all data points. Distributions are significantly different (***) as measured by a two-sample Kolmogorov-Smirnov test ($n = 70, 302$ selective outcomes for each rule out of 500; $p<10^{-10}$; $D = 0.45$). (**E**) The response of a single cortical cell to L-events of different sizes (color) as a function of the sliding threshold for the BCM rule with $H_{\mathrm{int}} = 3.5$ and $v_0 = 0.7$. The cell's incoming synaptic weights from presynaptic thalamic neurons undergo LTP or LTD depending on L-event size. (**F**) Probability of L-event size contributing to LTD (left) and LTP (right) for the BCM rule with the same parameters as in E.

The online version of this article includes the following figure supplement(s) for figure 3:

**Figure supplement 1.** Effect of initial bias on receptive field refinements and topography.

**Figure supplement 2.** Peripheral L-events generate robust receptive field refinement in the absence of H-events.

be generated over a wider range of H-inter-event-intervals $H_{\mathrm{int}}$ and target rates $v_0$ for the BCM rule (*Figure 3C*, right).

Despite this apparent advantage of the BCM rule, it generates receptive fields with much worse topography than the Hebbian covariance rule (*Figure 3D*). The underlying reason for this worse topography of the BCM rule is the sign of synaptic change evoked by L-events of different sizes corresponding to different participation rates. In particular, small L-events with low participation rates generate postsynaptic cortical activity smaller than the sliding threshold and promote long-term synaptic depression (LTD), while large L-events with high participation rates generate cortical activity larger than the sliding threshold and promote long-term synaptic potentiation (LTP) (*Figure 3E,F*).

Therefore, the amount of information for connectivity refinements present in the small L-events is limited in the BCM learning rule resulting in poor topographic organization of receptive fields.

Taken together, our results confirm that H-events can operate in parallel to network refinements by L-events and homeostatically regulate connection strength as postulated. However, the formation of receptive fields by the Hebbian covariance rule is very sensitive to small changes in event properties (e.g. inter-event intervals), which are common throughout development (*Rochefort et al., 2009*). In this case, H-events are disruptive and lead to the elimination of all thalamocortical synapses, effectively decoupling the cortex from the sensory periphery. In the BCM rule, including H-events prevents the decoupling of cortical cells from the periphery because the amount of LTD is dynamically regulated by the sliding threshold on cortical activity. However, L-events lose the ability to instruct topography because they generate LTP primarily when they are large. Therefore, neither learning rule seems suitable to organize network connectivity between the thalamus and cortex during development.

## Adaptive H-events achieve robust selectivity

After comparing the distinct outcomes of the Hebbian and BCM learning rules in the presence of L- and H-events, we proposed that a mechanism that regulates the amount of LTD during H-events based on cortical activity, similar to the sliding threshold of the BCM rule, could be a biologically plausible solution to mitigate the decoupling of cortical cells in the Hebbian covariance rule. This mechanism combined with the Hebbian learning rule could lead to refined receptive fields that also have good topographic organization. Hence, we postulated that H-events adapt by assuming that during H-events cortical cells scale their amplitude to the average amplitude of the preceding recent events. In particular, for each cortical cell $j$ an activity trace $\eta_j$ integrates the cell's firing rate $v_j$ over a timescale $\tau_\eta$ slower than the membrane time constant:

$$\tau_\eta \frac{\mathrm{d}\eta_j(t)}{\mathrm{d}t} = -\eta_j(t) + v_j(t). \tag{5}$$

This activity trace $\eta_j$ then scales the intrinsic firing rate of the cortical cells during an H-event, $H_{\mathrm{amp}} \rightarrow \eta_j H_{\mathrm{amp}}$, making it dependent on its recent activity. The activity trace $\eta_j$ might biophysically be implemented through a calcium-dependent signaling pathway that is activated upon sufficient burst depolarization and that is able to modulate a cell's excitability in the form of plasticity of intrinsic excitability (*Desai et al., 1999*; *Daoudal and Debanne, 2003*; *Tien and Kerschensteiner, 2018*). A fast, activity-dependent mechanism that decreases single-neuron excitability following a prolonged period of high network activity has been identified in spinal motor neurons of neonatal mice (*Lombardo et al., 2018*). However, there might be other ways to implement this adaptation (see Discussion).

Using adaptive H-events, we investigated the refinement of receptive fields in the network with the same Hebbian covariance rule (*Figure 4A*). In sharp contrast to the Hebbian covariance rule with non-adaptive H-events (*Figure 3A*), we observed that changing the average inter-event interval of H-events in a wider and more biologically realistic range (from the data, $H_{\mathrm{int}} \sim 3L_{\mathrm{int}}$) yields selectivity and appropriately refined receptive fields (*Figure 4A*). Increasing $\theta_u$ or decreasing $H_{\mathrm{int}}$ yields progressively smaller receptive fields while mitigating cortical decoupling (*Figure 4B*). The refined receptive fields also have a very good topography because L-events in the Hebbian learning rule carry nearest-neighbor information for the topographic refinements (*Figure 4C*). The proportion of selective receptive fields for adaptive H-events, however, is much higher than for their non-adaptive counterparts (390 vs. 70 out of 500 simulations). These results persist when the participation rates of L- and H-events change. For instance, when H-events involve 70–100% of cortical neurons, the percent of outcomes with selective receptive fields (75.0%) and the percent of outcomes with decoupled cortical neurons (0%) remain similar.

Next, we investigated how the proposed adaptive mechanism scales H-event amplitude by modulating the relative strength of H-events. For the Hebbian covariance rule, we calculated the analytical solution for weight development with L- and H-events by reducing the dimension of the system to two: one being the average of the weights that potentiate and form the receptive field, $w_{\mathrm{RF}}$, and the other being the average of the remaining weights, which we called 'complementary' to the receptive field, $w_{\mathrm{C}}$ (*Figure 4D*; see Appendix, Materials and methods). We calculated the phase

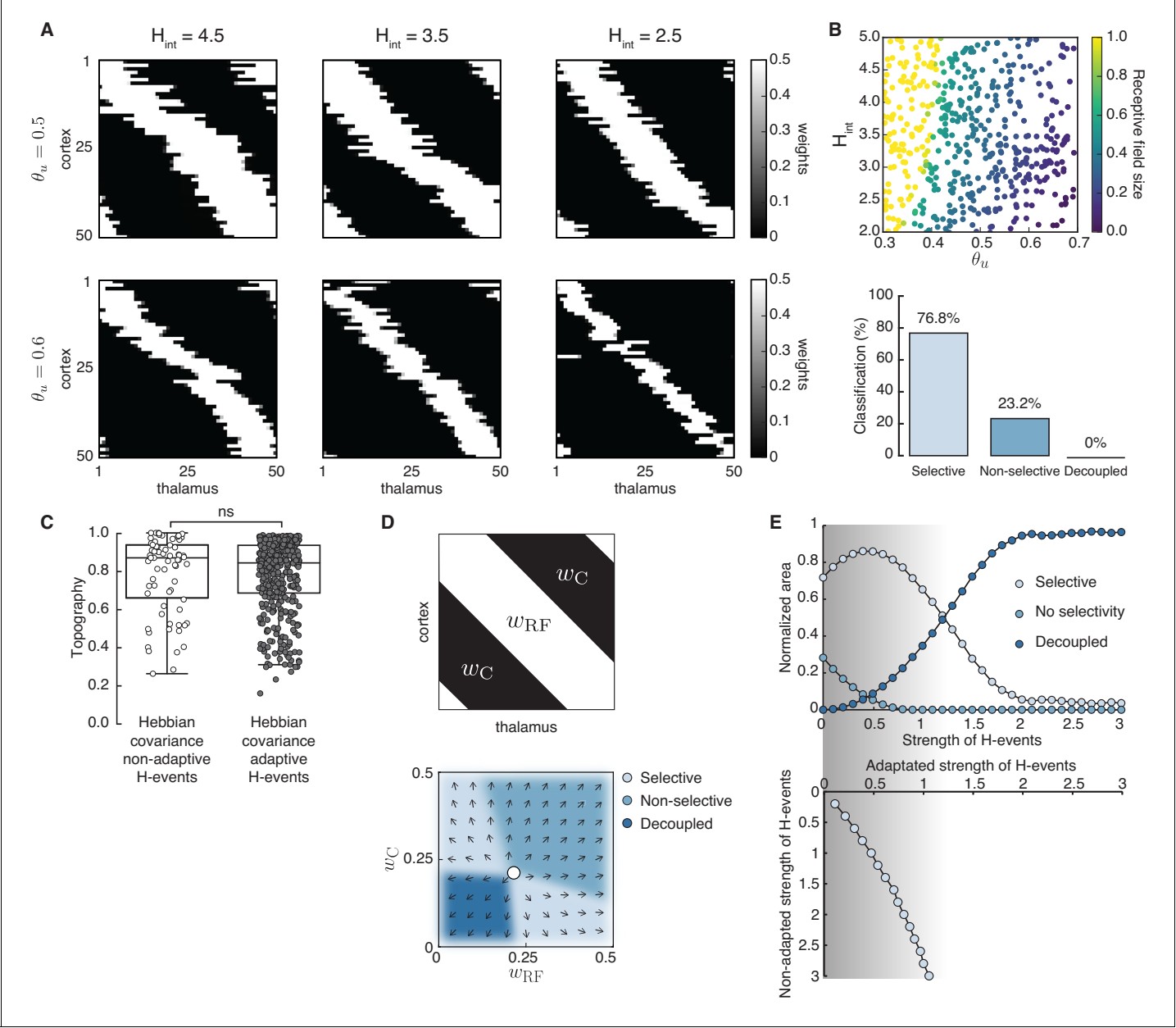

**Figure 4.** Adaptive cortical events refine thalamocortical connectivity. (A) Receptive field refinement with adaptive H-events and different H-inter-event-intervals, $H_{int}$. Top: $\theta_u = 0.5$; bottom: $\theta_u = 0.6$. (B) Receptive field sizes from 500 Monte Carlo simulations for combinations of $H_{int}$ and $\theta_u$. Bottom: Percentage of simulation outcomes classified as 'selective' when the average receptive field size is smaller than one and larger than 0, 'non-selective' when the average receptive field size is equal to 1, and 'decoupled' when the average receptive field size is 0 for the two rules. (C) Topography of receptive fields classified as selective in B. Horizontal line indicates median, the box is drawn between the 25th and 75th percentile, whiskers extend above and below the box to the most extreme data points that are within a distance to the box equal to 1.5 times the interquartile range and points indicate all data points. Distributions are not significantly different (ns) as measured by a two-sample Kolmogorov-Smirnov test ($n = 70, 390$ selective outcomes for each rule out of 500; $p = 0.41$; $D = 0.45$). (D) Top: Reduction of the full weight dynamics into two dimensions. Two sets of weights were averaged: those which potentiate and form the receptive field, $w_{RF}$, and the complementary set of weights that depress, $w_C$. Bottom: Initial conditions in the reduced two-dimensional phase plane were classified into three outcomes: 'selective', 'non-selective', and 'decoupled'. We sampled 2500 initial conditions which evolved according to *Equation 16* until the trajectories reached one of the selective fixed points, $(w_{max}, 0)$ and $(0, w_{max})$, or resulted in no selectivity either because both weights depressed to $(0, 0)$ or potentiated to $(w_{max}, w_{max})$. The normalized number of initial coordinates generating each region can be interpreted as the area of the phase plane that results in each outcome. (E) Top: Normalized area of the phase plane of the reduced two-dimensional system that resulted in 'selective', 'non-selective', and 'decoupled' outcomes for $\theta_u = 0.53$ as a function of H-event strength. The darker shading indicates ranges of non-adapted H-event strength where the selectivity area is maximized. Bottom: The corresponding adapted strength of H-events was calculated in simulations with adaptive H-events and plotted as a function of the nominal, non-adapted strength of H-events.

*Figure 4 continued on next page*

*Figure 4 continued*

The range of adapted H-event strengths (bottom) corresponds to the range of non-adaptive values that maximize the selectivity area (top). Each point shows the average over 10 runs and the bars the standard deviation (which are very small).

plane area of the reduced two-dimensional system with non-adaptive H-events (calculated as the proportion of initial conditions) that results in selectivity, potentiation or depression (*Figure 4D*, bottom). We found that adaptively modulating the strength of H-events maximizes the area of the phase plane that results in selectivity (*Figure 4E*, shaded region). The range of H-event strengths that maximizes the selective area for each input threshold in the reduced two-dimensional system can be related to the scaling of H-event amplitude in the simulations (Methods). In particular, the adaptation reliably shifts the H-event amplitude that would have occurred without adaptation, which we call 'non-adapted strength of H-events', into the regime of amplitudes that maximizes selectivity, which we call 'adapted strength of H-events' (*Figure 4E*). Therefore, the adaptation of H-event amplitudes controls the selective refinement by peripheral L-events by modulating the cortical depression by adapted H-events.

### *In vivo* spontaneous cortical activity shows a signature of adaptation

To determine whether spontaneous cortical activity contains a signature of our postulated adaptation mechanism of H-event amplitudes, we reanalyzed published *in vivo* two-photon $Ca^{2+}$ imaging data recorded in the visual cortex of young mice (P8-10) (*Siegel et al., 2012*). We combined multiple consecutive ~ 300 s long recordings for up to 40 mins of data from a given animal. First, we tested for long-term fluctuations in cortical excitability in the concatenated recordings of the same animal. We identified L- and H-events based on previously established criteria (*Siegel et al., 2012*). We found that the average amplitude of all (L and H) events is not significantly different across consecutive recordings of the same animal (*Figure 5—figure supplement 1A*). Additionally, across different animals and ages, individual event amplitudes remain uncorrelated between successive recordings at this timescale (*Figure 5—figure supplement 1B*). This suggests that there are no prominent long-term amplitude fluctuations, and therefore, the correlations cannot be explained by such fluctuations. Even so, slow amplitude fluctuations would not be able to generate refined receptive fields in the model (*Figure 5—figure supplement 2*).

Next, we investigated the relationship between the amplitude of a given H-event and the average activity preceding it. For each detected H-event, we extracted all spontaneous (L- or H-) events that preceded this H-event up to $T_{max} = 300$ s before it. We then scaled the amplitude of each previous event multiplying it by an exponential kernel with a decay time constant of $\tau_{decay} = 1000$ s, which is sufficiently long to integrate many preceding spontaneous events (compared with the inter-event intervals in *Figure 1B*), and averaged these scaled amplitudes to get an aggregate quantity over amplitude and frequency (see Materials and methods).

We found that this aggregate amplitude of L- and H-events preceding a given H-event is significantly correlated ($r = 0.44$, $p < 10^{-10}$) to the amplitude of the selected H-event (*Figure 5B*). Consequently, a strong (weak) H-event follows strong (weak) average preceding network activity (*Figure 5C*), suggesting that cortical cells adapt their spontaneous firing rates as a function of their previous activity levels. The correlations are robust to variations in the inclusion criteria, maximum time $T_{max}$ to integrate activity and the exponential decay time constant $\tau_{decay}$ (*Figure 5—figure supplement 3*).

### Modulating spontaneous activity properties affects receptive field refinements

Our results make relevant predictions for the refinement of receptive fields upon manipulating spontaneous activity. For example, H-event frequency can be experimentally reduced by a gap junction blocker (carbenoxolone) (*Siegel et al., 2012*). Our work demonstrates a trade-off between H-event frequency and the learning rule's threshold between potentiation and depression on receptive field size; hence, less frequent H-events will need a somewhat higher threshold to achieve the same receptive field size (*Figure 4*).

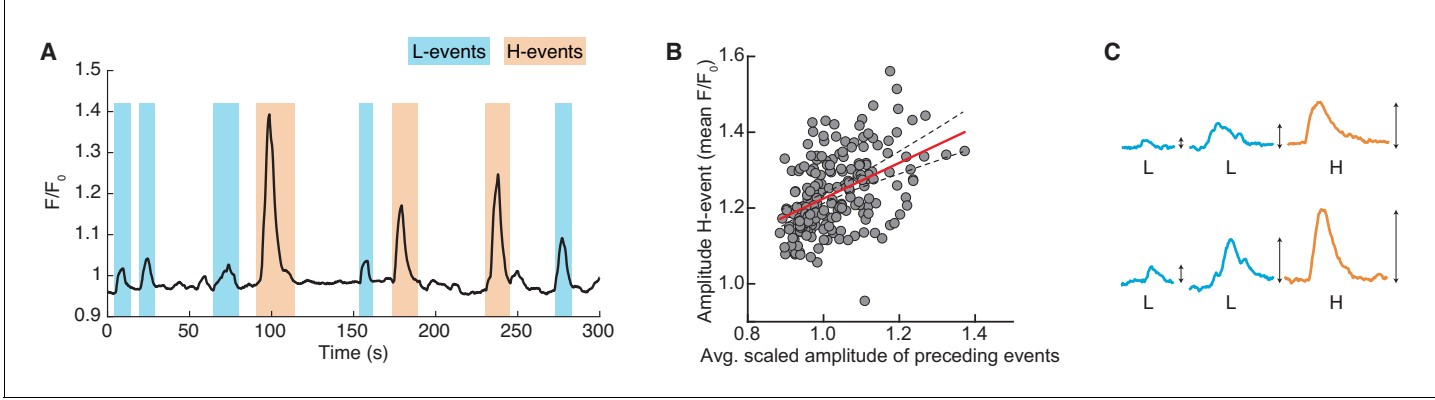

**Figure 5.** Spontaneous events in developing cortex adapt to recent activity. (A) Calcium trace of a representative recording with L- (blue) and H-events (orange) (*Siegel et al., 2012*). (B) The amplitude of an H-event shown as a function of the aggregate amplitude of preceding L- and H-events up to $T_{max} = 300$ s before it, scaled by an exponential kernel with a decay time constant of 1000 s ($N = 195$ events from nine animals). Animals with fewer than 12 H-events preceded by activity within $T_{max}$ were excluded from this analysis (see Materials and methods). The Pearson correlation coefficient is $r = 0.44$, $p < 10^{-10}$, $CI = (0.32, 0.54)$. Red line indicates regression line with 95% confidence bounds as dashed lines. (C) Schematic of the postulated adaptation: A weak (strong) H-event is more likely to be preceded by weak (strong) spontaneous events.

The online version of this article includes the following figure supplement(s) for figure 5:

**Figure supplement 1.** Fluctuations in cortical activity cannot generate correlations between event amplitude and average preceding activity.

**Figure supplement 2.** Long-term fluctuations in the amplitudes of L- and H-events cannot guide proper network refinement.

**Figure supplement 3.** Robustness of correlation under variations in the inclusion criteria, $T_{max}$ and $\tau_{decay}$.

Similarly, L-events can also be experimentally manipulated, for instance, by altering inhibitory signaling (*Leighton et al., 2020*), or the properties of retinal waves which propagate as L-events into the cortex. We performed Monte Carlo simulations with a range of input thresholds $\theta_u$ and variable participation rates of thalamic neurons in L-events, using the Hebbian covariance rule with adaptive H-events (*Figure 6A*). Larger L-events in our model produce less refined, that is, larger receptive fields in the cortical network (*Figure 6B,C*, left). This result is not surprising given the proposed role of L-events in guiding receptive field refinements, and is consistent with the imprecise and unrefined receptive fields observed in the visual cortex of animals where retinal wave properties have been modified. For instance, a prominent example of retinal wave manipulations are $\beta2$ knockout mice, which lack expression of the $\beta2$ subunit of the nicotinic acetylcholine receptor ($\beta2$-nAChR) that mediates spontaneous retinal waves in the first postnatal week. In these animals, retinal waves are consistently larger as characterized by the increased correlation with distance (*Sun et al., 2008*; *Stafford et al., 2009*; *Cutts and Eglen, 2014*), in addition to other features. As a result, there are measurable defects in the retinotopic map refinement of downstream targets (*Grubb et al., 2003*; *Cang et al., 2005b*; *Burbridge et al., 2014*). Smaller L-events also refine receptive fields with better topographic organization (*Figure 6C*, right) and do not impair connectivity refinements. This result could be linked to experiments where the expression of $\beta2$-nAChR is limited to the ganglion cell layer of the retina, resulting in smaller retinal waves than those in wild-type and undisturbed retinotopy in the superior colliculus (*Xu et al., 2011*), although the effects in the cortex are unknown.

Therefore, we suggest that certain manipulations that modulate the size of sensory activity from the periphery have a profound impact on the precision of receptive field refinement in downstream targets, making predictions to be tested experimentally. In contrast to retinal wave manipulations, the effect of altered inhibitory signaling on receptive field refinements is still unknown. It is likely that such manipulations will also affect H-events (*Leighton et al., 2020*), as well as shape ongoing plasticity in the network (*Wu et al., 2020*), and hence have less predictable effects on receptive field size and topography.

## Adaptive H-events promote the developmental event sparsification of cortical activity

Thus far, we focused on the development of the network connectivity in our model driven by spontaneous activity based on properties measured during a few postnatal days (P8-10, *Figure 4A*).

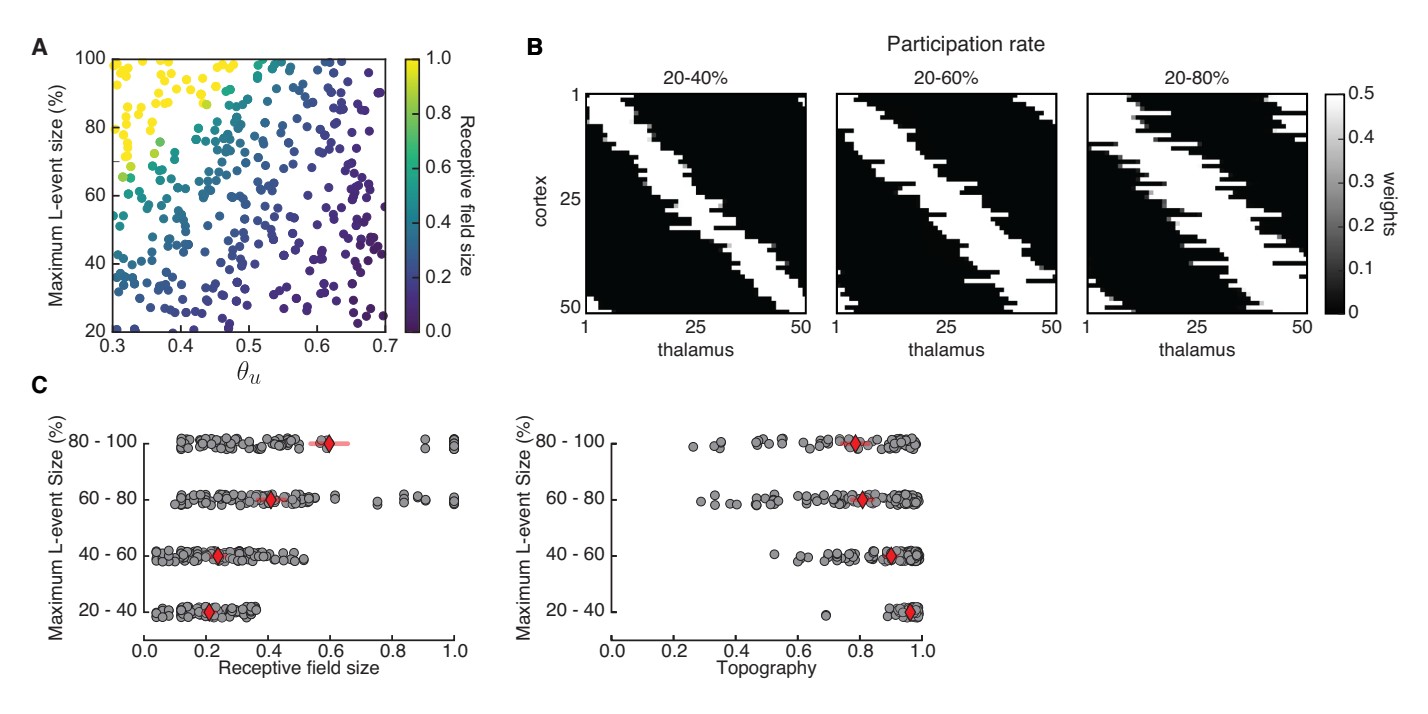

**Figure 6.** Receptive field refinement depends on the properties of L-events. (**A**) Receptive field sizes from 500 Monte Carlo simulations for different sizes of L-events where the minimum participation rate was 20%, and the maximum participation rate was varied. The input threshold was taken from the range $0.3 \leq \theta_u \leq 0.7$, while the adaptive H-events had a fixed inter-event-interval $H_{\text{int}} = 3.5$. (**B**) Individual receptive fields for different L-event maximum participation rates and $\theta_u = 0.50$. As the upper bound of the participation rate progressively increases from 40% to 80%, receptive fields get larger. (**C**) Left: Receptive field sizes from A binned according to the maximum L-event size. Right: Corresponding topography of selective receptive fields for different sizes of L-events. Diamonds in red indicate the mean, while horizontal bars indicate the 95% confidence interval.

However, *in vivo* spontaneous activity patterns are not static, but dynamically regulated during development by ongoing activity-dependent plasticity which continuously reshapes network connectivity that lasts several days (*Rochefort et al., 2009*; *Golshani et al., 2009*; *Frye and MacLean, 2016*). Moreover, it is unclear if the same criteria based on event participation rates and amplitude can be used to separate the spontaneous events into L and H at later developmental ages. Hence, we next asked how our observed modifications in network connectivity that are the result of receptive field refinement further modify spontaneous activity patterns on a much longer developmental timescale of several days in our model. Therefore, we analyzed all spontaneous events of simulated cortical neurons during the process of receptive field refinement in the presence of adaptive H-events (*Figure 4B*). Since the input threshold $\theta_u$ of the Hebbian learning rule is related to receptive field size (*Figure 4B*), we used $\theta_u$ as a proxy for time of development in the model: low $\theta_u$ corresponds to earlier developmental stages when receptive fields are large, while high $\theta_u$ corresponds to late developmental stages when receptive fields are refined. This assumption is also in line with the fact that the input resistance of neurons in V1 and S1 decreases during development (*Etherington and Williams, 2011*; *Golshani et al., 2009*), so that the depolarizing current necessary to trigger an action potential increases with age.

At an early developmental stage in the model ($\theta_u = 0.45$), the unrefined receptive fields of cortical neurons in our network model propagate thalamic activity into the cortex as very broad spontaneous events, while adaptive H-events remain intrinsic to the cortical layer. As in the data (*Figure 7A,C*; *Siegel et al., 2012*), the amplitude of events with 20–80% participation rate is approximately half the amplitude of events with greater than 80% participation rate (*Figure 7B,D*, left). Moreover, we also observed a high proportion of large events with greater than 80% participation rate (*Figure 7—figure supplement 1A*), suggesting that in the network model large spontaneous events are very frequent. At an intermediate developmental stage in the model ($\theta_u = 0.50$), as receptive fields refine

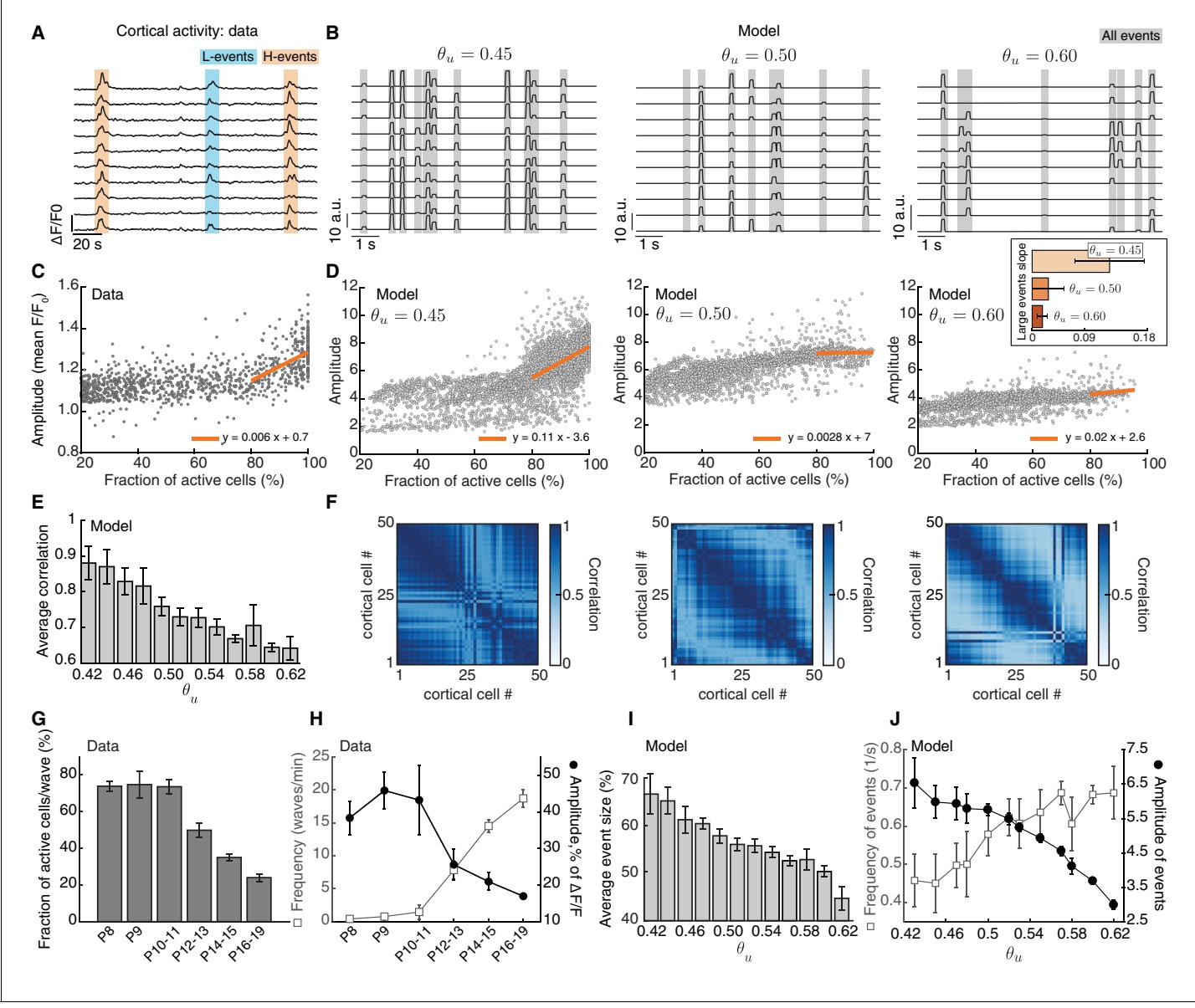

**Figure 7.** Adaptive H-events promote sparsification of cortical activity during development. (**A**) Spontaneous activity in the mouse visual cortex recorded *in vivo* at P8-10 (***Siegel et al., 2012***). Each activity trace represents an individual cortical cell. Blue and orange shading denotes L-events and H-events, respectively, as identified by ***Siegel et al., 2012***. (**B**) Sample traces of cortical activity for different values of $\theta_u$ (as a proxy for developmental age). Gray shading denotes all events detected in our networks. (**C**) Amplitude vs. participation rate plot from the data (***Siegel et al., 2012***). The regression line for the amplitude vs. participation rate in H-events has a positive slope. (**D**) Amplitude vs. participation rate plots from the model, for different values of $\theta_u$. Inset: The regression line for the amplitude vs. participation rate in large events with greater than 80% participation rate has a slope that decreases with $\theta_u$. Error bars represent standard deviation. (**E**) Correlation between cortical neurons decreases as a function of the input threshold $\theta_u$ in the model, a proxy for developmental time. (**F**) Correlation matrices of simulated cortical neuron activity corresponding to D. (**G,H**). Event sizes and the relationship between frequencies (open squares) and amplitudes (filled circles) of spontaneous events at different postnatal ages (data reproduced from ***Rochefort et al., 2009***). Error bars represent standard error of the mean (number of animals used at each age is provided in the original reference). (**I**) Spontaneous event sizes as a function of the input threshold $\theta_u$. (**J**) Frequencies (squares) and amplitudes (circles) of events with 20–80% participation rate in the model at different input thresholds. Error bars represent the standard error of the mean of 10 simulations.

The online version of this article includes the following figure supplement(s) for figure 7:

**Figure supplement 1.** Developmental event sparsification in the model.

and peripheral events activate fewer cortical neurons, our proposed adaptation of H-event amplitudes decreases the overall level of intrinsic activity in the cortical layer. This changes the relationship between effective amplitude and participation rate (*Figure 7D*, middle), with large events decreasing their amplitudes and density (*Figure 7—figure supplement 1A*). Finally, at late developmental stages in the model ($\theta_u = 0.60$), the relationship between effective amplitude and participation rate is almost absent (*Figure 7D*, right). Overall event amplitude is much lower resulting in far fewer large events with greater than 80% participation rate (*Figure 7—figure supplement 1B*). Therefore, due to the progressive receptive field refinements and the continued H-event adaptation in response to resulting activity changes, spontaneous events in our model progressively sparsify during ongoing development, whereby spontaneous events become smaller in size with fewer participating cells. This finding suggests that spontaneous events in the cortex at later developmental ages can no longer be separated into L and H using the same criteria of participation rate and amplitude as during early development. We also found that the mean pairwise correlation of all cortical neurons in the model decreases as a function of developmental age ($\theta_u$; *Figure 7E,F*), which further supports the trend of progressive sparsification already observed in the event sizes.

Interestingly, such event sparsification of spontaneous activity has been observed experimentally in the mouse barrel cortex during postnatal development from P4 to P26 (*Golshani et al., 2009*) and in the visual cortex from P8 to P79 (*Rochefort et al., 2009*). During this period, in the visual cortex, the size of spontaneous events decreases (*Figure 7G*), the amplitude of the participating cells also decreases, while event frequency increases (*Figure 7H*; *Rochefort et al., 2009*). This progressive event sparsification of cortical activity is generated by mechanisms intrinsic to the cortex, and does not seem to be sensory-driven (*Rochefort et al., 2009*; *Golshani et al., 2009*). We found the same relationships in our model using $\theta_u$ as a proxy for developmental time (*Figure 7I,J*).

In summary, our framework for activity-dependent plasticity and receptive field refinement between thalamus and cortex with adaptive H-events can tune the properties of cortical spontaneous activity and provide a substrate for the event sparsification of cortical activity during development on a much longer timescale than receptive field refinement. This sparsification has been found in different sensory cortices, including visual (*Rochefort et al., 2009*), somatosensory (*Golshani et al., 2009*), and auditory (*Frye and MacLean, 2016*), suggesting a general principle that underlies network refinement. However, the event sparsification we observe is different from sparse network activity implicated in sparse efficient coding, which interestingly seems to decrease during development (*Berkes et al., 2009*; *Berkes et al., 2011*). Our modeling predicts that cortical event sparsification is primarily due to the suppression of cortically-generated H-events in the Hebbian covariance rule, which switches cortical sensitivity to input from the sensory periphery after the onset of sensory experience.

## Discussion

We examined the information content of spontaneous activity for refining local microcircuit connectivity during early postnatal development. In contrast to classical works on activity-dependent refinements, which used mathematically convenient formulations of spontaneous activity (*Willshaw and von der Malsburg, 1976*; *Mackay and Miller, 1990*), we used spontaneous activity patterns characterized in the mouse visual cortex *in vivo* before the onset of vision (P8-10), which revealed its rich structure. Specifically, we explored the joint contribution of two distinct patterns of spontaneous activity recorded in the mouse visual cortex before the onset of vision, local (L-events) and global (H-events), on establishing topographically refined receptive fields between the thalamus and the cortex without decoupling in a model network with activity-dependent plasticity. Because of their spatially correlated activity, we proposed that peripherally generated L-events enable topographic refinement, while H-events regulate connection strength homeostatically. We investigated two Hebbian learning rules – the Hebbian covariance and the BCM rules – which use joint pre- and postsynaptic activity to trigger synaptic plasticity. First, we studied the Hebbian covariance rule that induces global synaptic depression in the presence of only postsynaptic activity (i.e. H-events). Second, we studied the BCM rule, which is known to establish the emergence of ocular dominance and orientation selectivity in the visual system. Although L-events successfully instruct topographic receptive field refinements in the Hebbian covariance rule, naively including H-events provides too much depression, eliminating selectivity in the network despite fine-tuning (*Figure 3*). In contrast, in the

BCM rule, H-events are indeed homeostatic, regulating the threshold between depression and potentiation. However, small L-events, which carry precise information for topographic connectivity refinements, mostly cause long-term depression in the synaptic weights and disrupt topography. Inspired by the sliding threshold in the BCM rule, we proposed a similar adaptive mechanism operating at the single-cell level in the Hebbian covariance rule. This mechanism regulates the amplitude of the cortically generated H-events according to the preceding average activity in the network to homeostatically balance local increases and decreases in activity, and can successfully refine receptive fields with excellent topography (*Figure 4*). Without any additional fine-tuning, this mechanism can also explain the long-term event sparsification of cortical activity as the circuit matures and starts responding to visual input (*Figure 7*). Therefore, we propose that L- and adaptive H-events cooperate to synergistically guide circuit organization of thalamocortical synapses during postnatal development.

## The origin of cortical event amplitude adaptation

After a re-examination of spontaneous activity recorded in the developing cortex *in vivo* between postnatal days 8 and 10 (*Siegel et al., 2012*), we found evidence for the proposed H-event amplitude adaptation (*Figure 5*). This mechanism is sufficiently general in its formulation that it could be realized at the cellular, synaptic or network level. At the cellular level, the adaptation mechanism resembles the plasticity of intrinsic excitability. Typically, plasticity of intrinsic excitability has been reported in response to long-term perturbations in activity or persistent changes in synaptic plasticity like LTP and LTD, where the intrinsic properties of single neurons are adjusted in an activity-dependent manner (*Daoudal and Debanne, 2003*; *Desai et al., 1999*). During plasticity of intrinsic excitability, neurons can alter the number and expression levels of ion channels to adjust their input-output function either by modifying their firing thresholds or response gains, which could represent the substrate for H-event amplitude regulation. Our adaptation mechanism is consistent with the fast plasticity of intrinsic excitability operating on the timescale of several spontaneous events supported by many experimental studies. For instance, intrinsic excitability of spine motoneurons is depressed after brief but sustained changes in spinal cord network activity in neonatal mice (*Lombardo et al., 2018*). Similarly, hippocampal pyramidal neurons also exhibit a rapid reduction of intrinsic excitability in response to sustained depolarizations lasting up to several minutes (*Sánchez-Aguilera et al., 2014*). In addition to reduced excitability, in the developing auditory system, enhanced intrinsic excitability has been reported in the cochlea followed by reduced synaptic excitatory input from hair cells in a model of deafness, although this change is slower than our proposed adaptation mechanism (*Babola et al., 2018*).

At the synaptic level, our adaptation mechanism can be implemented by synaptic scaling, a process whereby neurons regulate their activity by scaling incoming synaptic strengths in response to perturbations (*Turrigiano et al., 1998*). A second possibility is short-term depression, which appears to underlie the generation of spontaneous activity episodes in the chick developing spinal cord (*Tabak et al., 2001*; *Tabak et al., 2010*). Similarly, release probability suppression has been reported to strongly contribute to synaptic depression during weak activity at the calyx of Held (*Xu and Wu, 2005*), which is more pronounced at immature synapses where morphological development renders synaptic transmission less effective (*Renden et al., 2005*; *Nakamura and Takahashi, 2007*). This is also the case in the cortex, where short-term synaptic plasticity in young animals is stronger (*Oswald and Reyes, 2008*). Beyond chemical synapses, plasticity of gap junctions, which are particularly prevalent in development (*Niculescu and Lohmann, 2014*), could also be a contributing mechanism that adapts overall network activity (*Cachope et al., 2007*; *Haas et al., 2011*).

Finally, at the network level, the development of inhibition could be a substrate for amplitude adaptation of cortically generated events. The main inhibitory neurotransmitter, GABA, is thought to act as a depolarizing neurotransmitter, excitatory in early postnatal days (*Ben-Ari, 2002*), although recent evidence argues that GABAergic neurons have an inhibitory effect on the cortical network already in the second postnatal week (*Murata and Colonnese, 2020*; *Kirmse et al., 2015*; *Valeeva et al., 2016*). Thus, the local maturation of inhibitory neurons – of which there are several types (*Tremblay et al., 2016*) – that gradually evolve to balance excitation and achieve E/I balance (*Dorrn et al., 2010*) could provide an alternative implementation of the proposed H-event adaptation.

## Developmental sparsification of cortical activity

On a longer timescale than receptive field refinement, we demonstrated that the adaptation of H-event amplitude can also bring about the event sparsification of cortical activity, as global, cortically generated H-events are attenuated and become more localized (*Figure 7*). The notion of 'sparse neural activity' has received significant attention in experimental and theoretical studies of sensory processing in the cortex, including differing definitions and implementations (*Field, 1994*; *Willmore and Tolhurst, 2001*; *Berkes et al., 2009*; *Olshausen and Field, 2004*; *Zylberberg and DeWeese, 2013*). In particular, sparse activity in the mature cortex has been argued to be important for the efficient coding of sensory inputs of different modalities (*Olshausen and Field, 2004*; *Field, 1994*). Hence, the developmental process of receptive field refinement might be expected to produce sparser network activity over time. However, experiments directly testing this idea have found no, or even opposite, evidence for the developmental emergence of efficient sparse coding (*Berkes et al., 2009*; *Berkes et al., 2011*). In the context of our work, sparsification simply refers to an overall sparsification of network events (fewer active cells per event). Given that our data pertain to developmental spontaneous activity before eye-opening, in complete absence of stimulation, it is not straightforward to relate our event sparsification to the sparse efficient coding hypothesis.

## Assumptions in the model

Our model is based on the assumption that L- and H-events have distinct roles during the development of the visual system. Retinal waves, the source of L-events, carry information downstream about the position and function of individual retinal ganglion cells (*Stafford et al., 2009*), hence they are ideally suited to serve as 'training patterns' to enable activity-dependent refinements based on spatiotemporal correlations (*Ko et al., 2011*; *Ackman and Crair, 2014*; *Thompson et al., 2017*). Since all cells are maximally active during H-events, these patterns likely do not carry much information that can be used for activity-dependent refinement of connectivity. In contrast, we assumed that H-events homeostatically control synaptic weights, operating in parallel to network refinements by L-events (*Figure 4*). Indeed, highly correlated network activity can cause homeostatic down-regulation of synaptic weights via a process known as synaptic scaling (*Turrigiano and Nelson, 2004*). The homeostatic role of H-events is also consistent with synaptic downscaling driven by slow waves during sleep, a specific form of synchronous network activity (*Tononi and Cirelli, 2006*; *Vyazovskiy et al., 2008*). Since during development sleep patterns are not yet regular, we reasoned that refinement (by L-events) and homeostasis (by H-events) occur simultaneously instead of being separated into wake and sleep states.

We focused on the role of spontaneous activity in driving receptive field refinements rather than study how spontaneous activity is generated. While the statistical properties of spontaneous activity in the developing cortex are well-characterized, the cellular and network mechanisms generating this activity remain elusive. In particular, while H-event generation has been shown to rely on gap junctions (*Siegel et al., 2012*; *Niculescu and Lohmann, 2014*), which recurrently connect developing cortical cells, not much is known about how the size of cortical events is modulated and how an L-event is prevented from spreading and turning into an H-event. It is likely that cortical inhibition plays a critical role in localizing cortical activity and shaping receptive field refinements (*Wood et al., 2017*; *Leighton et al., 2020*), for instance, through the plasticity of inhibitory connections by regulating E/I balance (*Dorrn et al., 2010*). As new experiments are revealing more information about the cellular and synaptic mechanisms that generate spatiotemporally patterned spontaneous activity (*Fujimoto et al., 2019*), a full model of the generation and the effect of spontaneous activity might soon be feasible.

The threshold parameter in the Hebbian covariance rule in the presence of H-events implements an effective subtractive normalization that sharpens receptive fields (see Appendix). Despite the strong weight competition, subtractive normalization seems to be insufficient to stabilize receptive fields in the presence of non-adaptive H-events (*Figure 3*). Multiplicative normalization is an alternative normalization scheme, but it does not generate refined receptive fields (*Miller and MacKay, 1994*). Therefore, we also studied the BCM rule due to its ability to generate selectivity in postsynaptic neurons under patterned input. While the BCM rule successfully generates selectivity and receptive field refinement, the resulting topography is worse than in the Hebbian covariance rule (*Figure 3*). Both rules have an adaptive component: in the BCM rule it is the threshold between

potentiation and depression that slides as a function of postsynaptic activity, while in the Hebbian covariance rule it is the adaptive amplitude of H-events, while the rule itself is fixed. Although experiments have shown the stereotypical activity dependence of the BCM rule (*Kirkwood et al., 1996*; *Sjöström et al., 2001*), whether a sliding threshold for potentiation vs. depression exists is still debated. Moreover, the timescale over which the threshold slides to prevent unbounded synaptic growth needs to be much faster than found experimentally (*Zenke et al., 2017*). Our proposed H-event amplitude adaptation operates on the fast timescale of several spontaneous events found experimentally (*Siegel et al., 2012*; *Sánchez-Aguilera et al., 2014*; *Lombardo et al., 2018*). Hence, together with the better topography and the resulting event sparsification as a function of developmental stage that the Hebbian covariance rule with adaptive H-events generates, we propose it as the more likely plasticity mechanism to refine receptive fields in the developing visual cortex.

Finally, we have focused here on the traditional view that molecular gradients set up a coarse map that activity-dependent mechanisms then refine (*Goodhill and Xu, 2005*). In our model, this was implemented as a weak bias in the initial connectivity, which did not affect our results regarding the refinement of receptive fields. Both activity and molecular gradients may work together in interesting ways to refine receptive fields (*Grimbert and Cang, 2012*; *Godfrey and Swindale, 2014*; *Naoki, 2017*), and future work should include both aspects.

## Predictions of the model

Our model makes several experimentally testable predictions. First, we showed that changing the frequency of H-events can affect the size of the resulting receptive fields under both the BCM (*Figure 3*) and the Hebbian covariance rule with adaptive H-events (*Figure 4*). The frequency of H-events can be experimentally manipulated using optogenetics or pharmacology. For instance, gap junction blocker (carbenoxolone) has been shown to specifically reduce the frequency of H-events (*Siegel et al., 2012*), hence in that scenario our results predict broader receptive fields.

Additionally, L-events can also be experimentally manipulated. Recently, reduced inhibitory signaling by suppressing somatostatin-positive interneurons have has been shown to increase the size of L-events in the developing visual cortex (*Leighton et al., 2020*). With the effect of altered inhibitory signaling on receptive field refinements still unknown, our work predicts larger receptive fields and worse topography upon reduction of inhibition. L-events can also be experimentally manipulated by changing the properties of retinal waves, which can significantly affect retinotopic map refinement of downstream targets (*Grubb et al., 2003*; *Cang et al., 2005b*; *Burbridge et al., 2014*). Indeed, $\beta 2$ knockout mice discussed earlier have larger retinal waves and less refined receptive fields in the visual cortex (*Sun et al., 2008*; *Stafford et al., 2009*; *Cutts and Eglen, 2014*). If we assume that these larger retinal waves manifest as larger L-events in the visual cortex following *Siegel et al., 2012*, then these experimental observations are in agreement with our model results.

Third, our model predicts that as a result of receptive field refinement during development, network events sparsify as global, cortically generated events are attenuated and become more localized. Interestingly, the properties of spontaneous activity measured experimentally in different sensory cortices (*Rochefort et al., 2009*; *Frye and MacLean, 2016*; *Smith et al., 2015*; *Ikezoe et al., 2012*; *Shen and Colonnese, 2016*; *Golshani et al., 2009*) and in the olfactory bulb (*Fujimoto et al., 2019*) change following a very similar timeline during development as predicted in our model. However, in many of these studies activity has not been segregated into peripherally driven L-events and cortically generated H-events. Therefore, our model predicts that the frequency of L-events would increase while the frequency of H-events would decrease over development.

Finally, we propose that for a Hebbian covariance rule to drive developmental refinements of receptive fields using spontaneous L- and H-event patterns recorded *in vivo* (*Siegel et al., 2012*), H-events need to adapt to ongoing network activity. Whether a fast adaptation mechanism like the one we propose operates in the cortex requires prolonged and detailed activity recordings *in vivo*, which are within reach of modern technology (*Ackman and Crair, 2014*; *Ji, 2017*; *Gribizis et al., 2019*). Our framework also predicts that manipulations that affect overall activity levels of the network, such as activity reduction by eye enucleation, would correspondingly affect the amplitude of ongoing H-events.

## Conclusion

In summary, we studied the refinement of receptive fields in a developing cortex network model constrained by realistic patterns of experimentally recorded spontaneous activity. We proposed that adaptation of the amplitude of cortically generated spontaneous events achieves this refinement without additional assumptions on the type of plasticity in the network. Our model further predicts how cortical networks could transition from supporting highly synchronous activity modules in early development to sparser peripherally driven activity suppressing local amplification, which could be useful for preventing hyper-excitability and epilepsy in adulthood while enhancing the processing of sensory stimuli.

# Materials and methods

## Network model

We studied a feedforward, rate-based network with two one-dimensional layers, one of $N_u$ thalamic neurons (*u*) and the other of $N_v$ cortical neurons (*v*), with periodic boundary conditions in each layer to avoid edge effects. The initial connectivity in the network was all-to-all with uniformly distributed weights in the range $w_{\mathrm{ini}} = [a, b]$. In addition, a topographic bias was introduced by modifying the initially random connectivity matrix to have the strongest connections between neurons at the matched topographic location, and which decay with a Gaussian profile with increasing distance (*Figure 2C*), with amplitude $s$ and spread $\sigma_s$. During the evolution of the weights, soft bounds were applied on the interval $[0, w_{\mathrm{max}}]$. We studied weight evolution under two activity-dependent learning rules: the Hebbian (*Equation 2*) and the BCM (*Equation 3*) rules. *Table 1* lists all parameters. Sample codes can be found at github.com/comp-neural-circuits/LH-events (*Wosniack, 2021*; copy archived at swh:1:rev:b90e189a9e1a4d0cdda097d435fa91b1236f1866).

## Generation of L- and H-events

We modeled two types of spontaneous events in the thalamic (L-events) and the cortical (H-events) layer of our model (*Siegel et al., 2012*). During L-events, the firing rates of a fraction ($L_{\mathrm{pct}}$) of neighboring thalamic neurons were set to $L_{\mathrm{amp}} = 1$ during a period $L_{\mathrm{dur}}$ and were otherwise 0. Similarly, during H-events, the firing rates of a fraction ($H_{\mathrm{pct}}$) of cortical cells were set to $H_{\mathrm{amp}}$ during time $H_{\mathrm{dur}}$. As a result, cortical neuron activity was composed of H-events and L-events transmitted from the thalamus. For each H-event, $H_{\mathrm{amp}}$ was independently sampled from a Gaussian distribution with mean $H_{\mathrm{amp}}$ and standard deviation $H_{\mathrm{amp}}/3$. The inter-event intervals were $L_{\mathrm{int}}$ and $H_{\mathrm{int}}$ sampled from experimentally characterized distributions in *Siegel et al., 2012*, (*Table 1*). The event durations and inter-event intervals were shortened by a factor of 10 compared to the values measured in the data (*Figure 1*) to speed up our simulations, but the relationships observed in the data were preserved. We note that in the experiments, both L- and H-events were characterized in the primary visual cortex; in our model, we assume that L-events are generated in the retina and subsequently propagated through the thalamus to the cortex, where they manifest with the experimentally reported characteristics (see *Figure 7B* for example). This interpretation is supported by experimental evidence (*Siegel et al., 2012*), but we cannot exclude the possibility that the retina also generates H-events or that L-events are generated in the cortex.

## Reduction of the weight dynamics to two dimensions

To reduce the full weight dynamics to a two-dimensional system, we averaged all the $n$ weights belonging to the receptive field that are predicted to potentiate along the initial topological bias, as $w_{\mathrm{RF}}$, and all the $N_u - n$ remaining weights, which we call complementary to the receptive field, as $w_{\mathrm{C}}$. When all weights behaved the same, we arbitrarily split them into two groups of the same size. Details about the classification of weights as $w_{\mathrm{RF}}$ or $w_{\mathrm{C}}$ can be found in the Appendix.

## Computing the strength of simulated H-events

To relate the reduced two-dimensional phase planes to the simulation results, we wrote down the steady state activity of neuron $j$ (*Equation 1*), which contains the rate gain from H-events relative to L-events, $\langle R_{\mathrm{H}} \rangle$ (also called 'Strength of H-events' in *Figure 4E*):

$$\langle R_{\mathrm{H}} \rangle = \frac{\langle L_{\mathrm{int}} \rangle}{\langle H_{\mathrm{int}} \rangle} \frac{\langle H_{\mathrm{amp}} \rangle}{\langle L_{\mathrm{amp}} \rangle} \frac{\langle H_{\mathrm{dur}} \rangle}{\langle L_{\mathrm{dur}} \rangle} = \frac{L_{\mathrm{int}}}{H_{\mathrm{int}}} \langle H_{\mathrm{amp}} \rangle \tag{6}$$

since $\langle L_{\mathrm{dur}} \rangle = \langle H_{\mathrm{dur}} \rangle$ and $\langle L_{\mathrm{amp}} \rangle = 1$.

In the absence of adaptive H-events, for a fixed set of values for $H_{\mathrm{amp}}$ and $L_{\mathrm{int}}$ (as in **Table 1**) and a chosen $\langle R_{\mathrm{H}} \rangle$ which we called 'Non-adapted strength of H-events' in **Figure 4E**, we used **Equation 6** to find the $H_{\mathrm{int}}$ value that satisfies the equation. Next, we ran simulations with the same $H_{\mathrm{int}}$ and $L_{\mathrm{int}}$ parameters, but adaptive $H_{\mathrm{amp}}$. We fixed the inter-event intervals of both L- and H-events to their mean values $L_{\mathrm{int}}$ and $H_{\mathrm{int}}$ instead of sampling them from distributions in **Figure 4E**. Then we numerically estimated the average amplitude of H-events with adaptation, which we called 'Adapted strength of H-events' in **Figure 4E** at the end of the simulation (final 5% of the simulation time) when the dynamics were stationary.

## Receptive field statistics

The following receptive field statistics were used to quantify properties of the weight matrix $W$ after the developing weights became stable.

### Receptive field size

The receptive field of a cortical neuron is the group of weights from thalamic cells for which $w_{ij} > w_{\mathrm{max}}/5$. The lower threshold was chosen to make the measurement robust to small fluctuations around 0, which are present because of the soft bounds. Mathematically, we compute the receptive field size of cortical neuron $j$ as:

$$R_F(\boldsymbol{w}_j) = \frac{1}{N_u} \sum_{i=1}^{N_u} \mathcal{I}_i, \tag{7}$$

with the $\boldsymbol{\mathcal{I}}$ vector given by:

$$\mathcal{I}_i = \begin{cases} 1, & w_{ij} > w_{max}/5 \\ 0, & \text{otherwise.} \end{cases} \tag{8}$$

The normalized receptive field ranges from 0 corresponding to a total decoupling of the cortical cell from the input layer, to one corresponding to no selectivity due to the potentiation of all weights from the input layer to that neuron. To compute the average receptive field size of the network, we include only the cortical neurons ($N^*$) that have not decoupled:

$$R_F(\boldsymbol{W}) = \frac{1}{N^*} \sum_{j=1}^{N^*} R_F(\boldsymbol{w}_j). \tag{9}$$

If all the cortical cells have decoupled from the thalamus, we set $R_F(\boldsymbol{W}) = 0$.

### Topography $\mathcal{T}$

The topography of the network is a measure of how much of the initially weak biased topography is preserved in the final receptive field. Due to our biased initial conditions, neighboring thalamic cells are expected to project to neighboring cortical cells, yielding a diagonal weight matrix. For each cortical neuron, we calculated how far the center of its receptive field is from the ideal diagonal. Mathematically, for each row $j$ of $W$, we determined the center of the receptive field $c_j$ and calculated the smallest distance (while considering periodic boundary conditions) between the receptive field center and the diagonal element $j$. Then, we summed all the squared distances and calculated the average error of the topography:

$$\xi = \frac{1}{N_v} \sum_{j=1}^{N_v} |c_j - j|^2. \tag{10}$$

To normalize the topography, we compared $\xi$ to the topography error $\Xi$ of a column receptive field (**Figure 3—figure supplement 1A**) where the centers of all cortical receptive fields were the

same, $c_j = c$ ($c$ a constant). For such a column receptive field, $\Xi = \frac{N_u^2}{12}$. Therefore, we define the topography score $\mathcal{T}$ as:

$$\mathcal{T} = 1 - \frac{\xi}{\Xi}. \tag{11}$$

The topography will be close to one if the weight matrix is perfectly diagonal and 0 if the final receptive field is a column ($\xi = \Xi$).

### Proportion of cortical decoupling $\mathcal{D}$

To quantify the cortical decoupling, we use *Equation 8* to compute the fraction of decoupled neurons divided by the number of neurons, $\frac{1}{N_u}\sum_{j=1}^{N_u}\prod_{i=1}^{N_u}(1 - \mathcal{I}_{ij})$. If the decoupling is 0, no cortical neuron has decoupled from the thalamus, while decoupling of 1 means that all the cortical neurons are decoupled from the thalamus.

## Quantifying adaptation in the data

We first investigated if fluctuations in the activity across recordings could generate significant correlations. We analyzed consecutive recordings (each ~5 mins long) in the same animal of which we had between 3 and 14 in all 26 animals (separated by <5 mins due to experimental constraints on data collection) to identify possible fluctuations on a longer timescale. We found that the average amplitude of all (L and H) events is not significantly different across consecutive recordings of the same animal (*Figure 5—figure supplement 1A*, one-way ANOVA tests, p>0.05 in 23 out of 26 animals). Across different animals and ages, individual event amplitudes remained uncorrelated between successive recordings at this timescale, which we confirmed by plotting the difference in event amplitude as a function of the time between recordings (*Figure 5—figure supplement 1B*), Kruskal-Wallis test, p>0.05.

For our reanalysis of the spontaneous events (*Figure 5*), we only included events that recruited at least 20% of the cells in the imaging field of view following *Siegel et al., 2012*. We computed the average amplitude of all events that occurred within a time window $T_{max}$ before an H-event (consecutive recordings were concatenated) and compared it to the amplitude of the H-event. We excluded animals that had fewer than 12 H-events preceded by spontaneous activity within the time window $T_{max}$ (nine animals remained after exclusion). Next, we computed the correlation coefficient of the relationship between H-event amplitude and the average amplitude of preceding activity within $T_{max}$ with a leaky accumulator of time constant $\tau_{decay}$. To estimate the 95% confidence interval, we performed a bootstrap analysis in which we generated 1000 bootstrap datasets by drawing without replacement from the valid pairs of H-event amplitudes and average amplitude of preceding activity. We repeated this analysis with different thresholds for excluding data (*Figure 5—figure supplement 3A,B*), different values of the time window $T_{max}$ within which events are averaged (*Figure 5—figure supplement 3C*) and for different decay time constants $\tau_{decay}$ (*Figure 5—figure supplement 3D*). All data and analysis code can be found at github.com/comp-neural-circuits/LH-events.

## Spontaneous events identification in the model

To quantify the properties of spontaneous activity in the cortical layer of our model, we used the time series of activity of all the simulated cortical neurons (after weight stabilization is achieved) sampled in a high time resolution (0.01 s, *Figure 7B*). We defined a global activity threshold $\nu = v_{max}/r$, where $v_{max}$ is the highest amplitude among the cortical cells in the recording and $r$ is a fixed scaling constant ($r = 8$ for all recordings). For each cortical cell $j$, we labeled the intervals where the cell was active (1) or inactive (0) based on:

$$x_j(t) = \begin{cases} 1, & \text{if } v_j(t) \geq \nu, \\ 0, & \text{otherwise.} \end{cases} \tag{12}$$

We then used the trace $X(t) = \sum_{j=1}^{N} x_j(t)$ to define the number of active cortical cells at each time step $t$, that is, the participation rate. For each identified event, we averaged the amplitude of the active cells to obtain the amplitude vs. participation rate relationship.

## Acknowledgements

This project has received funding from the European Research Council (ERC) under the European Union's Horizon 2020 research and innovation programme (Grant agreement No. 804824 to JG). This work was further supported by the Max Planck Society (MW, JHK, LYC, JG), a NARSAD Young Investigator Grant from the Brain and Behavior Research Foundation (JG), a Capes-Humboldt Research fellowship (MW), the Smart Start joint training program in computational neuroscience (JHK), and by grants of the Netherlands Organization for Scientific Research (NWO, ALW Open Program grants, no. 819.02.017, 822.02.006 and ALWOP.216; ALW Vici, no. 865.12.001) and the "Stichting Vrienden van het Herseninstituut" (NZ, CL). We thank Stephen Eglen for discussions and ideas in the initial stages of the project.

## Additional information

### Funding

| Funder | Grant reference number | Author |
| --- | --- | --- |
| Alexander von Humboldt-Stiftung | | Marina E Wosniack |
| H2020 European Research Council | 804824 | Jan H Kirchner<br>Julijana Gjorgjieva |
| Max-Planck-Gesellschaft | | Marina E Wosniack<br>Jan H Kirchner<br>Ling-Ya Chao<br>Julijana Gjorgjieva |
| Brain and Behavior Research Foundation | 26253 | Julijana Gjorgjieva |
| SMART START training program in computational neuroscience | | Jan H Kirchner |
| Nederlandse Organisatie voor Wetenschappelijk Onderzoek | 819.02.017 | Nawal Zabouri<br>Christian Lohmann |
| Stichting Vrienden van het Herseninstituut | 805254845 | Nawal Zabouri<br>Christian Lohmann |
| Nederlandse Organisatie voor Wetenschappelijk Onderzoek | 822.02.006 | Nawal Zabouri<br>Christian Lohmann |
| Nederlandse Organisatie voor Wetenschappelijk Onderzoek | ALWOP.216 | Nawal Zabouri<br>Christian Lohmann |
| Nederlandse Organisatie voor Wetenschappelijk Onderzoek | ALW Vici no. 865.12.001 | Nawal Zabouri<br>Christian Lohmann |

The funders had no role in study design, data collection and interpretation, or the decision to submit the work for publication.

### Author contributions

Marina E Wosniack, Conceptualization, Software, Formal analysis, Methodology, Writing - original draft, Writing - review and editing, Performing simulations; Jan H Kirchner, Methodology, Writing - original draft; Ling-Ya Chao, Software, Methodology; Nawal Zabouri, Data curation, Investigation; Christian Lohmann, Conceptualization, Resources, Writing - review and editing; Julijana Gjorgjieva, Conceptualization, Resources, Supervision, Funding acquisition, Methodology, Writing - original draft, Project administration, Writing - review and editing

### Author ORCIDs

Marina E Wosniack (iD) https://orcid.org/0000-0003-2175-9713
Jan H Kirchner (iD) https://orcid.org/0000-0002-9126-0558
Ling-Ya Chao (iD) https://orcid.org/0000-0002-6706-5939

Christian Lohmann [ORCID] https://orcid.org/0000-0002-1780-2419
Julijana Gjorgjieva [ORCID] https://orcid.org/0000-0001-7118-4079

**Decision letter and Author response**
Decision letter https://doi.org/10.7554/eLife.61619.sa1
Author response https://doi.org/10.7554/eLife.61619.sa2

## Additional files

### Supplementary files

• Transparent reporting form

### Data availability

Sample codes and data are available at https://github.com/comp-neural-circuits/LH-events (copy archived at https://archive.softwareheritage.org/swh:1:rev:b90e189a9e1a4d0cdda097d435-fa91b1236f1866/).

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

## Appendix 1

### The weight dynamics under the Hebbian covariance rule

Since synaptic plasticity operates on a much slower time scale than the response dynamics of the output neuron, we make a steady state assumption and write *Equation 1* for neuron $j$ as:

$$v_j(t) = \boldsymbol{u}^\top(t)\boldsymbol{w}_j(t) + \langle R_{\mathrm{H}}\rangle(t), \tag{13}$$

where $\boldsymbol{w}_j$ is the vector of the elements in the row of the weight matrix $\boldsymbol{W}$, that is, the vector of weights from all thalamic neurons into cortical neuron $j$. $\langle R_{\mathrm{H}}\rangle(t)$ is the rate gain from H-events relative to L-events, which depends on the duration, amplitude, and inter-event intervals of both L- and H-events. Specifically, $\langle R_{\mathrm{H}}\rangle$ is proportional to $\langle H_{\mathrm{amp}}\rangle$, $\langle H_{\mathrm{dur}}\rangle$, $\langle L_{\mathrm{int}}\rangle$ and inversely proportional to $\langle H_{\mathrm{int}}\rangle$, $\langle L_{\mathrm{amp}}\rangle$ and $\langle L_{\mathrm{dur}}\rangle$ (where $\langle\cdot\rangle$ denotes an ensemble average over the activity patterns), such that:

$$\langle R_{\mathrm{H}}\rangle = \frac{\langle L_{\mathrm{int}}\rangle}{\langle H_{\mathrm{int}}\rangle} \frac{\langle H_{\mathrm{amp}}\rangle}{\langle L_{\mathrm{amp}}\rangle} \frac{\langle H_{\mathrm{dur}}\rangle}{\langle L_{\mathrm{dur}}\rangle}. \tag{14}$$

Rewriting *Equation 2* in vector form:

$$\tau_w \frac{\mathrm{d}\boldsymbol{w}_j(t)}{\mathrm{d}t} = \langle v_j(t)(\boldsymbol{u}(t) - \theta_u)\rangle. \tag{15}$$

Inserting *Equation 13* into the weight dynamics from *Equation 15* yields (dependence on $t$ is omitted for clarity):

$$\begin{aligned} \tau_w \frac{\mathrm{d}\boldsymbol{w}_j}{\mathrm{d}t} &= \langle (\boldsymbol{u}^\top\boldsymbol{w}_j + R_{\mathrm{H}})(\boldsymbol{u} - \theta_u)\rangle \\ &= \boldsymbol{C}\boldsymbol{w}_j - (\theta_u - \langle\boldsymbol{u}\rangle)\langle\boldsymbol{u}^\top\rangle\boldsymbol{w}_j - \langle R_{\mathrm{H}}\rangle\theta_u \\ &= \boldsymbol{C}'(\theta_u)\boldsymbol{w}_j - \langle R_{\mathrm{H}}\rangle\theta_u, \end{aligned} \tag{16}$$

where $\boldsymbol{C} = \boldsymbol{Q} - \langle u\rangle^2$ is the input covariance matrix, $\boldsymbol{Q} = \langle\boldsymbol{u}\boldsymbol{u}^\top\rangle$ is the input correlation matrix and we have defined $\boldsymbol{C}'(\theta_u) = \boldsymbol{C} - \langle u\rangle(\theta_u - \langle u\rangle) = \boldsymbol{Q} - \langle u\rangle\theta_u$ to be the 'modified covariance matrix' of the learning rule. We write $\langle\boldsymbol{u}\rangle$ to denote the vector with repeated element $\langle u\rangle$, which is the mean normalized size of an L-event (e.g. for L-events engaging 20–80% of input neurons, $\langle u\rangle = 0.5$). We also used the fact that the occurrence of L- and H-events is uncorrelated (*Siegel et al., 2012*), such that $\langle R_{\mathrm{H}}\boldsymbol{u}\rangle = 0$.

### Normalization constraints

The unconstrained Hebbian covariance rule has the undesirable effect that all the synaptic inputs to an output cell are potentiated and no selectivity is achieved. However, the presence of the threshold and H-events in *Equation 16* effectively implements a subtractive constraint in the weight dynamics (*Miller and MacKay, 1994*). In general, a subtractive constraint can be written as:

$$\tau_w \frac{\mathrm{d}\boldsymbol{w}_j}{\mathrm{d}t} = \boldsymbol{A}\boldsymbol{w}_j - \varepsilon(\boldsymbol{w}_j)\boldsymbol{n}, \tag{17}$$

where $\boldsymbol{A}$ is a symmetric matrix, $\boldsymbol{w}_j$ the weight vector, $\varepsilon(\boldsymbol{w}_j)$ a scalar function of the weights and $\boldsymbol{n}$ a constant vector of ones. By comparing the terms of the second line of *Equation 16*, we can identify $\boldsymbol{C} = \boldsymbol{A}$, since the covariance matrix is symmetric, and the function $\varepsilon(\boldsymbol{w}_j)$ as:

$$\varepsilon(\boldsymbol{w}_j) = (\theta_u - \langle u\rangle)\langle\boldsymbol{u}^\top\rangle\boldsymbol{w}_j - \langle R_{\mathrm{H}}\rangle\theta_u. \tag{18}$$

Due to the subtractive constraint in *Equation 16*, the synaptic weights will saturate at either the upper or lower bounds, resulting in the sharpening of receptive fields. Notice that if $\langle R_{\mathrm{H}}\rangle = 0$, the decay term in the subtractive constraint is proportional to $\boldsymbol{w}_j(t)$. Then, the subtractive component can be adjusted to induce less depression and prevent the decoupling of cortical cells. With non-adaptive H-events, however, $\langle R_{\mathrm{H}}\rangle \neq 0$ adds a constant decay rate to the weight dynamics that is independent of weight strength and therefore decouples cortical cells.

## The input correlation matrix

The spontaneous L-events in our network model have useful mathematical properties that allowed us to derive an analytical expression for the elements of the input correlation matrix $\boldsymbol{Q}$. The locality and periodicity of L-events and the long-term averaging of their stationary dynamics generate a symmetric and circulant matrix $\boldsymbol{Q}$ (*Gray, 2005*). In a circulant matrix, each row is rotated one element to the right relative to the preceding row, and thus $\boldsymbol{Q}$ can be completely defined by a vector.

Let $\ell$ be the fixed size of an L-event between 0 and $N_u$. By computing the correlations among all the possible L-events of size $\ell$, we can write the elements of the vector $q_{\ell k} = [q_{\ell 1}, q_{\ell 2}, \ldots, q_{\ell N_u}]$, which completely defines $\boldsymbol{Q}$, as:

$$q_{\ell k} = \frac{\ell - \min(\min(N_u - k - 1, k - 1), \min(N_u - \ell, \ell))}{N_u}, \tag{19}$$

where $\min(a,b)$ returns the minimum of $a$ and $b$. Since we wanted to explore the dependence of refinement on the size of L-events, defined by the minimum ($\ell_{\min}$) and maximum ($\ell_{\max}$) number of cells they activate in the input layer, we average over the size of a given event:

$$q_k = \frac{1}{(\ell_{\max} - \ell_{\min})} \sum_{\ell = \ell_{\min}}^{\ell_{\max}} \frac{\ell - \min(\min(N_u - k - 1, k - 1), \min(N_u - \ell, \ell))}{N_u}. \tag{20}$$

Finally, using the emergent symmetry of $\boldsymbol{Q}$, we can simplify the elements of this vector as follows:

$$q_k = q - \min(k - 1, N_u - k + 1)\delta_q, \quad \text{for all} \quad k = 1, 2, \ldots, N_u, \tag{21}$$

with $q = (\ell_{\max} + \ell_{\min})/2N_u$ and $\delta_q = 1/N_u$.

Since $\boldsymbol{C}(\theta_u)$ and $\boldsymbol{C}'(\theta_u)$ are defined in terms of $\boldsymbol{Q}$ minus a constant, both are circulant matrices as well.

## Fixed point of the weight dynamics

The fixed point of the weight dynamics under L- and H-events is obtained by setting *Equation 16* to zero and solving the resulting equation for $\boldsymbol{w}^*$:

$$\boldsymbol{w}^* = \langle R_{\mathrm{H}} \rangle \theta_u \boldsymbol{C}'(\theta_u)^{-1} \boldsymbol{n}, \tag{22}$$

where $\boldsymbol{n}$ is a vector of ones. To study the nature of the fixed point $\boldsymbol{w}^*$, we need to investigate the eigenvalues of the circulant matrix $\boldsymbol{C}'(\theta_u)^{-1}$ (the inverse of a circulant matrix is also a circulant matrix). A circulant matrix has the property that its eigenvectors can be written in terms of roots of unity. The eigenvalues are real and can be written as the discrete Fourier transforms of any row of the matrix (*Gray, 2005*). In particular, one of the eigenvalues is the sum of the elements of any row of the matrix, with a corresponding eigenvector that is a constant. We call this special eigenvalue the 'row-sum eigenvalue'. All eigenvalues except for the row-sum are non-negative.

We can write the fixed point as $\boldsymbol{w}^* = (\lambda^*)^{-1}\theta_u\langle R_{\mathrm{H}}\rangle\boldsymbol{n}$, where $\lambda^*$ is the row-sum eigenvalue of $\boldsymbol{C}'(\theta_u)$. It is clear that if only L-events are present, $\langle R_{\mathrm{H}} \rangle = 0$, the fixed point is always at the origin, $\boldsymbol{w}^* = 0$. Consequently, adding H-events in the cortical layer moves the location of the fixed point along the diagonal of the phase plane, that is, equally in all directions. The fixed point will be positive (negative) when $\lambda^*$ is positive (negative). Furthermore, it will be an unstable fixed point for $\lambda^* > 0$, since all the eigenvalues of the dynamical system are positive. The fixed point is a saddle node when $\lambda^* < 0$ since the other eigenvalues are positive.

## Calculation of receptive field size

The Hebbian covariance rule on its own does not have a mechanism to prevent the weights to grow infinitely large or negative. Thus, to generate receptive fields we imposed a lower bound at 0 and an upper bound at $w_{max}$. This enabled us to calculate the size of the receptive field, $n$, as a function of L-event properties and the input threshold, $\theta_u$ in the absence of H-events. Because the cortical cells are not recurrently connected, here we examine the weight vector onto a single cortical neuron. Assuming the dynamical system has reached steady state, to get a receptive field of size $n < N_u$ for

this cortical neuron, we can write the fixed point weight vector as $\boldsymbol{w}^* = (w_{\max}, \ldots, w_{\max}, 0, \ldots, 0)^\top$, where $n$ of the weights have reached the upper bound $w_{max}$, while the remaining weights the lower bound 0. The specific weight identity reaching the upper bound is not important, as long as they are topographically near each other. To achieve this fixed point:

$$\dot{w}_i|_{\boldsymbol{w}=\boldsymbol{w}^*} = \begin{cases} \geq 0, & 1 \leq i \leq n \\ < 0, & i > n. \end{cases} \tag{23}$$

Using the structure of our circulant correlation matrix (*Equation 21*), we can rewrite *Equation 16* (note that in the absence of H-events $\langle R_H \rangle = 0$) as:

$$\tau_w \begin{bmatrix} \dot{w}_1 \\ \vdots \\ \dot{w}_n \\ \dot{w}_{n+1} \\ \vdots \\ \dot{w}_{N_u} \end{bmatrix} = \begin{bmatrix} q_1 - \langle u \rangle \theta_u & \cdots & q_n - \langle u \rangle \theta_u & q_{n+1} - \langle u \rangle \theta_u & \cdots & q_{N_u} - \langle u \rangle \theta_u \\ \vdots & & \vdots & \vdots & & \vdots \\ q_n - \langle u \rangle \theta_u & \cdots & q_1 - \langle u \rangle \theta_u & q_2 - \langle u \rangle \theta_u & \cdots & q_{n+1} - \langle u \rangle \theta_u \\ q_{n+1} - \langle u \rangle \theta_u & \cdots & q_2 - \langle u \rangle \theta_u & q_1 - \langle u \rangle \theta_u & \cdots & q_{n+2} - \langle u \rangle \theta_u \\ \vdots & & \vdots & \vdots & & \vdots \\ q_2 - \langle u \rangle \theta_u & \cdots & q_{n+1} - \langle u \rangle \theta_u & q_{n+2} - \langle u \rangle \theta_u & \cdots & q_1 - \langle u \rangle \theta_u \end{bmatrix} \begin{bmatrix} w_{\max} \\ \vdots \\ w_{\max} \\ 0 \\ \vdots \\ 0 \end{bmatrix}. \tag{24}$$

Now, we study the conditions to guarantee that *Equation 23* is satisfied. Computing each equation explicitly, we obtain:

$$\tau_w \dot{w}_1 = (q_1 + \ldots + q_n) w_{\max} - n \langle u \rangle \theta_u w_{\max}$$
$$\vdots$$
$$\tau_w \dot{w}_n = (q_n + \ldots + q_1) w_{\max} - n \langle u \rangle \theta_u w_{\max}$$
$$\tau_w \dot{w}_{n+1} = (q_{n+1} + \ldots + q_2) w_{\max} - n \langle u \rangle \theta_u w_{\max} \tag{25}$$
$$\vdots$$
$$\tau_w \dot{w}_{N_u} = (q_2 + \ldots + q_{n+1}) w_{\max} - n \langle u \rangle \theta_u w_{\max}.$$

And finally, using *Equation 21* and after some manipulations this can be written as:

$$\tau_u \dot{w}_i = \left\{ nq - \left[ \frac{i(i-1)}{2} + \frac{(n-i)(n-i+1)}{2} \right] \delta_q - n \langle u \rangle \theta_u \right\} w_{\max}, \quad i = 1, 2, \ldots, N_u. \tag{26}$$

To determine the input threshold $\theta_u$ in *Equation 25* that yields a receptive field of size $n$, we assume that $\dot{w}_i|_{\boldsymbol{w}=\boldsymbol{w}^*} = 0$ for $i = n$; this implies that $\dot{w}_i|_{\boldsymbol{w}=\boldsymbol{w}^*} > 0$ for $i < n$ and $\dot{w}_i|_{\boldsymbol{w}=\boldsymbol{w}^*} < 0$ for $i > n$. We write $\theta_u$ as a linear combination of $q$ and $\delta_q$:

$$\theta_u = \alpha q + \beta \delta_q. \tag{27}$$

Using this ansatz in *Equation 26* and setting it to zero for $i = n$ (since $w_{\max} > 0$), we find that $\alpha = 1/\langle u \rangle$ and $\beta = -(n-1)/(2\langle u \rangle)$. Therefore, only in the presence of L-events, we derive the input threshold at which the resulting receptive field size is $n$:

$$\theta_u^n = \frac{q - \frac{(n-1)}{2} \delta_q}{\langle u \rangle}. \tag{28}$$

Plugging this threshold into *Equation 26* for all $i$, we get a quadratic polynomial in $i$:

$$\tau_w \dot{w}_i = \left( -i^2 + (n+1)i - n \right) \delta_q w_{\max}, \quad i = 1, 2, \ldots, N_u. \tag{29}$$

Since $\delta_q, w_{\max} > 0$, indeed $1 \leq i \leq n$ results in $\dot{w}_i \geq 0$ while $i > n$ yields $\dot{w}_i < 0$, thus, satisfying *Equation 23*.

In the absence of H-events, we computed the average size of receptive fields using *Equation 28* for a range of input thresholds $\theta_u$ and maximum participation rates of L-events, while keeping the minimum participation rate at 20% (*Appendix 1–figure 1A*, contour lines). We verified our analytical predictions with Monte Carlo simulations for the same range of parameters (*Appendix 1–figure 1A*). We confirmed that the size of L-events, which depends on the range of participation rates, has

a direct impact on receptive field size, with larger L-events resulting in larger receptive fields for a fixed input threshold (*Appendix 1—figure 1B*). Low input thresholds generate refined receptive fields only if the size of spontaneous events is small.

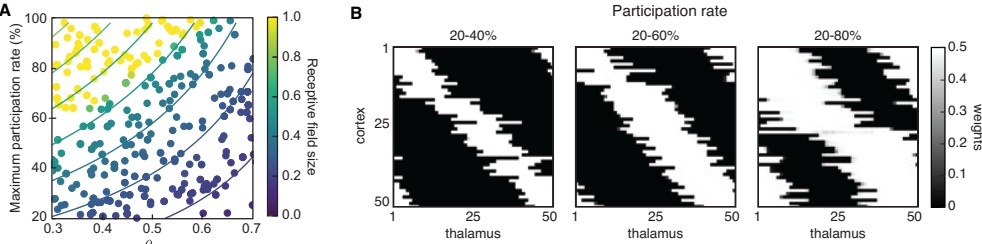

**Appendix 1—figure 1.** Receptive field size depends on L-event properties and learning rule input threshold in the absence of H-events. (**A**) Receptive field sizes from 500 Monte Carlo simulations for combinations of L-event maximum participation rate and input threshold, $\theta_u$. For all simulations, the L-event minimum participation rate was fixed at 20%. The contour plots of receptive field sizes were obtained using the analytical approach (*Equation 23*). (**B**) Example receptive fields for different L-event maximum participation rates and $\theta_u = 0.5$. Smaller events recruiting only 20–40% of the input neurons generate very refined receptive fields. As the upper bound of the participation rate progressively increases from 40% to 80%, receptive fields get larger.

## The eigenspace of $C'(\theta_u)$

To gain intuition for the weight dynamics in *Equation 16*, we first investigated the eigenspace of $C'(\theta_u)$, the vector space spanned by the eigenvectors of $C'(\theta_u)$. Specifically, we focused on the conditions that enabled the robust formation of cortical receptive fields.

Using the fact that $Q$ and $C'(\theta_u)$ are circulant matrices (*Appendix 1—figure 2A, B*), we identified two input thresholds, $\theta^*$ and $\theta^{**}$, that define three dynamical regions that the row-sum eigenvalue of $C'(\theta_u)$, $\lambda^*$, can occupy (*Appendix 1—figure 2C*). To obtain the first critical input threshold $\theta^*$ that characterizes the transition from region (i) to region (ii), we set, for any row $j$, the row-sum eigenvalue to the largest (fixed) eigenvalue of $C'(\theta_u)$:

$$\sum_{i=1}^{N_u} Q_{ji} - N_u \theta^* \langle u \rangle = \lambda_1,  \tag{30}$$

and for the input statistics of experimentally measured L-events (*Table 1*), we obtained $\theta^* = 0.414$. Similarly, the transition from region (ii) to region (iii) is achieved when the row-sum eigenvalue is set to zero:

$$\sum_{i=1}^{N_u} Q_{ji} - N_u \theta^{**} \langle u \rangle = 0,  \tag{31}$$

and the second critical input threshold is obtained as $\theta^{**} = 0.564$.

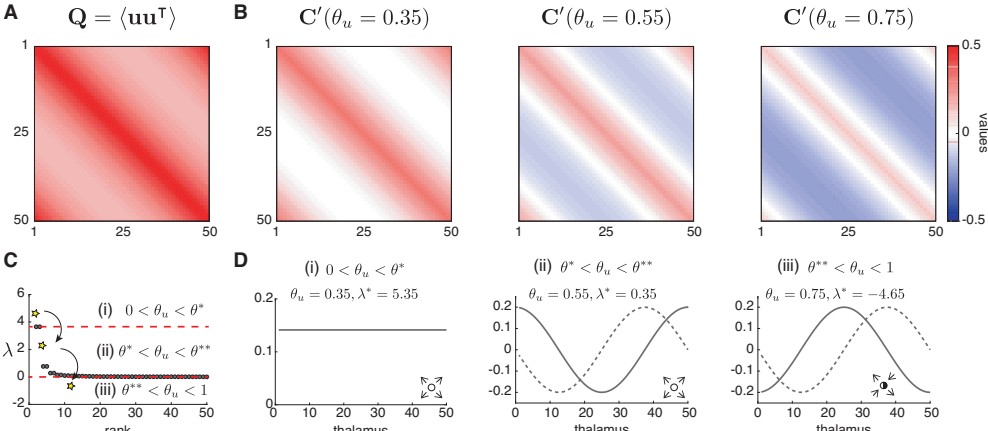

**Appendix 1—figure 2.** Eigenvalues and eigenvectors of weight dynamics predict receptive field refinement. (**A**) The input correlation matrix $Q = \langle uu^\top \rangle$. (**B**) The modified covariance matrices $C'(\theta_u) = Q - \langle u \rangle \theta_u$ for different input thresholds, $\theta_u$. (**C**) Two thresholds, $\theta^* = 0.414$ and $\theta^{**} = 0.564$, define three different dynamical regions in the spectrum of $C'(\theta_u)$, delineated by the horizontal red dashed lines: (i) $0 < \theta_u < \theta^*$, (ii) $\theta^* < \theta_u < \theta^{**}$, and (iii) $\theta^{**} < \theta_u < 1$. The row-sum eigenvalue $\lambda^*$ in each case is given by the yellow star, while the remaining eigenvalues are shown as gray circles. (**D**) Dominant eigenvectors corresponding to each region in C. Inset: fixed points corresponding to each region: (i), (ii) unstable node (open circles); (iii) saddle node (half-open circle).

In region (i), $0 < \theta_u < \theta^*$ and $\lambda^* > 0$ is the dominant (largest) eigenvalue of $C'(\theta_u)$. The eigenvector corresponding to it is a constant (*Appendix 1–figure 2D*, (i)). This predicts that all synaptic weights to a given cortical cell will potentiate preventing the formation of a localized receptive field. Since all eigenvalues are non-negative, the fixed point is an unstable node of the linear dynamical system (*Equation 16*).

In region (ii), $\theta^* < \theta_u < \theta^{**}$ and $\lambda^* > 0$. However, $\lambda^*$ is no longer the dominant eigenvalue. In this setting, there is a pair of dominant eigenvalues with the corresponding eigenvectors taking the form of out-of-sync sine waves with positive and negative elements. The sign of these elements predicts that some weights will potentiate while others depress, thus enabling the formation of receptive fields (*Appendix 1–figure 2D*, (ii)). All eigenvalues in this second region remain non-negative and the fixed point is still an unstable node of the dynamical system.

Finally, in region (iii), $\theta^{**} < \theta_u < 1$ and $\lambda^* < 0$. While the dominant eigenvectors are similar to those in region (ii), enabling the formation of localized receptive fields (*Appendix 1–figure 2D*, (iii)), the dynamics of the dynamical system are different because the fixed point is now a saddle node.

## Analysis of the weight dynamics in two dimensions

We reduced the dimension of the weight dynamics by defining two distinct sets of weights. The first set, $w_{\mathrm{RF}}$, corresponds to the $n$ weights which correspond to the topographically biased locations of the receptive field. The complementary set $w_{\mathrm{C}}$ contains the remaining weights. To classify the weights into $w_{\mathrm{RF}}$ and $w_{\mathrm{C}}$, first we solved *Equation 16* with the biased initial condition and limited the sum to the dominant eigenvalues and respective eigenvectors. In the case of selectivity, the eigenvectors have half the elements positive and half negative. Therefore, if the receptive field size is $n < N_u/2$ we needed to subsample the potentiating weights to achieve the smaller receptive field size. We did it by keeping only the $n$ largest positive elements in $w_{\mathrm{RF}}$ and moving the remaining ones to $w_{\mathrm{C}}$. If $n > N_u/2$, we downsampled $w_{\mathrm{C}}$ by moving the $n - N_u/2$ less-negative weights to $w_{\mathrm{RF}}$. Due to the topographically biased initial conditions, $w_{\mathrm{RF}}$ always contained the weights potentiating along the diagonal $w_{\mathrm{RF}} = w_{\mathrm{C}}$.

We then regularly sampled initial conditions in $[0, w_{\max}] \times [0, w_{\max}]$. For a given receptive field size $n$, we set $\theta_u = \theta_u^n$ according to *Equation 27*. If $w_{\mathrm{RF}}(0) > w_{\mathrm{C}}(0)$, $w_{\mathrm{RF}}$ contains the $n$ weights that form the receptive field by potentiating to the upper bound $w_{max}$, while $w_{\mathrm{C}}$ contains the remaining $N_u - n$ weights that depress to 0. Similarly, if $w_{\mathrm{RF}}(0) < w_{\mathrm{C}}(0)$, $w_{\mathrm{C}}$ contains the $n$ weights that potentiate to the upper bound, while $w_{\mathrm{RF}}$ contains the remaining $N_u - n$ weights that depress to 0. At each initial

condition, we solved the weight evolution (*Equation 16*) for each weight, and averaged the weights in $w_{\mathrm{RF}}$ and $w_{\mathrm{C}}$ for a small time interval to obtain the direction of the phase plane arrows. We computed the evolution trajectory by solving *Equation 16* with a topographically biased initial condition.

## Analytical solution of the two-dimensional weight dynamics with only L-events in the Hebbian covariance rule

We first examined the weight dynamics in the reduced two-dimensional phase plane $w_{\mathrm{RF}} \times w_{\mathrm{C}}$ for only L-events ($\langle R_{\mathrm{H}} \rangle = 0$ in *Equation 16*). The phase plane is symmetric about the diagonal $w_{\mathrm{RF}} = w_{\mathrm{C}}$ due to the symmetry of the dominant eigenvectors (*Appendix 1–figure 3A*). As predicted by the eigenvectors (*Appendix 1–figure 2D*), in region (i) both $w_{\mathrm{RF}}$ and $w_{\mathrm{C}}$ converge to the upper bound and the fixed point, which is located in the origin, is an unstable node (*Appendix 1–figure 3A*, left). Therefore, all weights potentiate and no receptive field can be formed. In regions (ii) and (iii), the eigenvectors predict the formation of receptive fields with $w_{\mathrm{RF}} \to w_{\mathrm{max}}$ and $w_{\mathrm{C}} \to 0$, respectively (*Appendix 1–figure 3A*, middle and right), although the dynamics are different in each case because the origin is an unstable node or a saddle node, respectively.

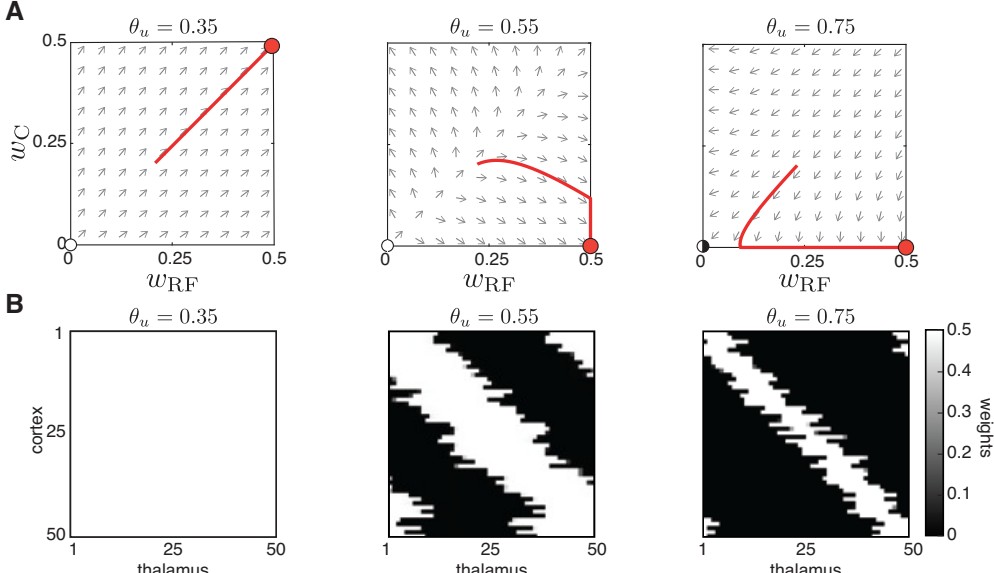

**Appendix 1—figure 3.** Peripheral L-events generate robust receptive field refinement. (**A**) The reduced two-dimensional weight dynamics in the phase plane with the same $\theta_u$ as *Appendix 1–figure 2B and D*. For each plane, the red trajectory depicts the weight evolution from an initial condition where $w_{\mathrm{RF}}(0) > w_{\mathrm{C}}(0)$ until the weights' upper bound ($w_{\mathrm{max}} = 0.5$). Left: $w_{\mathrm{C}} \to w_{\mathrm{max}}$ and $w_{\mathrm{RF}} \to w_{\mathrm{max}}$, resulting in no selectivity. Middle and right: $w_{\mathrm{C}} \to 0$ and $w_{\mathrm{RF}} \to w_{\mathrm{max}}$, resulting in selectivity and receptive field refinement. (**B**) Simulation results for the same input thresholds of *Appendix 1–figure 2B,D* for the full 50-dimensional system.

Our analytical predictions of the reduced two-dimensional system with only L-events were confirmed in numerical simulations of the full *N*-dimensional system. In particular, in region (i) all weights potentiate with each cortical cell receiving input from all thalamic inputs, such that no receptive field forms (*Appendix 1–figure 3B*, left). In regions (ii) and (iii), receptive fields form with good topography (*Appendix 1–figure 3B*, middle and right). Therefore, consistent with the analytical prediction of receptive field size (*Appendix 1–figure 1*), higher input thresholds resulted in smaller receptive fields.

Thus, the Hebbian covariance rule can generate receptive fields of size depending on the input threshold $\theta_u$ in the presence of only L-events originating from the sensory periphery. This result is in agreement with previous findings of the emergence of other aspects of development including topographic maps (*Willshaw and von der Malsburg, 1976*) and ocular dominance (*Miller et al., 1989*; *Miller, 1994*) or other selectivity (*Mackay and Miller, 1990*; *Miller and MacKay, 1994*; *Lee et al., 2002*) in the presence of correlated activity in the input layer of similar feedforward

networks. We find that when the only input to the cortex are peripheral L-events, intrinsic properties of the learning rule, such as the threshold between potentiation and depression, control receptive field refinement.

## Analytical solution of the two-dimensional weight dynamics with L- and H-events in the Hebbian covariance rule

We also studied how the addition of H-events affects network refinements. To investigate the role of H-events in a systematic way, we repeated our analytical study of the weight-dynamics from *Equation 16*, but with $\langle R_H \rangle \neq 0$. In the reduced two-dimensional phase plane, $w_{RF} \times w_C$, including spontaneous events in the cortical layer moves the fixed point of *Equation 16* away from the origin to the coordinates $w_{RF} = w_C = (\lambda^*)^{-1} \theta_u \langle R_H \rangle$. Nevertheless, the different dynamical regimes reported in n *Appendix 1–figure 2C* continue to be valid. In region (i), the addition of cortical events moves the unstable node away from the origin and into the first quadrant (*Appendix 1–figure 4A* top). As a result, a small region of selectivity emerges in the plane in which initial conditions generate refined receptive fields (*Appendix 1–figure 4A*, top middle). Therefore, the addition of H-events enables the emergence of weight selectivity but through a different mechanism than the one obtained with only L-events. Rather than modulating the learning rule through the input threshold, changing the H-event statistics through the $\langle R_H \rangle$ parameter can generate different receptive field sizes for a fixed input threshold. However, the strength of H-events has to be fine-tuned to generate refined receptive fields. Within a small range of $\langle R_H \rangle$ the network transitions from no-selectivity (where all weights potentiate, *Appendix 1–figure 4A*, top left) to complete decoupling (where all weights depress, *Appendix 1–figure 4A*, top right).

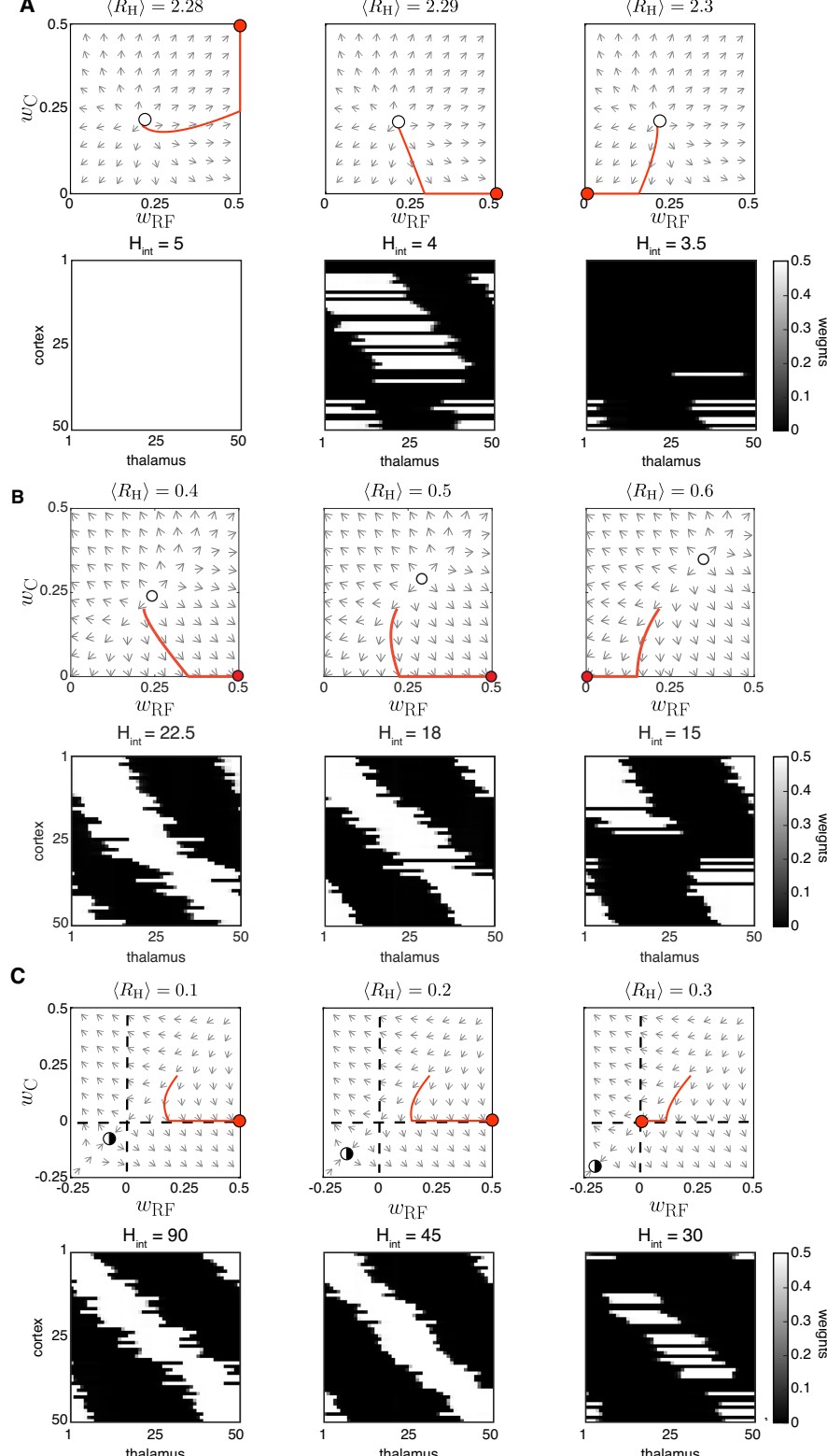

**Appendix 1—figure 4.** Spontaneous cortical H-events disrupt receptive field refinement. (**A**) Top: Phase planes of the reduced two-dimensional system for input threshold $\theta_u = 0.4$ (region i) and increasing strength of cortical events $\langle R_H \rangle$ with an example trajectory (red). Selectivity can only be observed for fine-tuned $\langle R_H \rangle$. The fixed point (open circle), an unstable node, has moved to the first

*Appendix 1—figure 4 continued on next page*

*Appendix 1—figure 4 continued*

quadrant. Bottom: Simulations of receptive field development with the same parameters where $H_{\text{int}}$ was progressively reduced (this is the same set of parameters as shown in *Figure 3A*). (**B**) Top: Phase planes for $\theta_u = 0.52$ (region ii) and increasing $\langle R_{\text{H}} \rangle$ with an example trajectory (red). The unstable node (open circle) moves from the origin to the first quadrant as $\langle R_{\text{H}} \rangle$ increases. Bottom. Simulations with the same parameters where $H_{\text{int}}$ was progressively reduced. (**C**) Top: Phase planes for $\theta_u = 0.6$ (region iii) show the transition from selective receptive fields to cortical decoupling in response to increasing $\langle R_{\text{H}} \rangle$. The fixed point (half-open circle), now a saddle node because $\lambda^* < 0$, has moved away from the origin to the third quadrant. Bottom: Simulations with the same parameters with very infrequent H-events where $H_{\text{int}}$ was progressively reduced.

To relate the reduced two-dimensional phase planes to the simulation results, we used *Equation 14* to obtain $\langle R_{\text{H}} \rangle$ by taking into account the simulation parameters in *Table 1*. We next verified the predictions of the reduced two-dimensional system in numerical simulations of the full network with H-events (*Appendix 1–figure 4A* bottom). To capture the gradual increase of $\langle R_{\text{H}} \rangle$ as in the reduced two-dimensional system, we decreased the average inter-event interval between H-events, $H_{\text{int}}$. As before, only a narrow range of $H_{\text{int}}$ leads to refined receptive fields, albeit with some degree of decoupling (*Appendix 1–figure 4A*, bottom middle). Outside of this range, individual cortical neurons are either non-selective (*Appendix 1–figure 4A*, bottom left) or nearly completely decoupled from the thalamus (*Appendix 1–figure 4A*, bottom right).

In regions (ii) and (iii), the fixed point moves from the origin to the first and third quadrants, respectively (*Appendix 1–figure 4B* and *Appendix 1–figure 4C*). In both cases, only very weak H-events can sustain finite receptive fields because the high input threshold value already provides sufficient depression to the network. We confirmed our analytical results in regions (ii) and (iii) with numerical simulations of the full network (*Appendix 1–figure 4B* and *Appendix 1–figure 4C*, bottom).

