## [Decision Letter]

**Acceptance summary:**

This computational modeling study examines how experimentally characterized patterns of spontaneous activity, thought to arise either locally or to be driven by events in the eye, can influence the developmental refinement of connectivity and receptive field properties in the visual cortex. The theoretical framework set out by the authors provides new insight into the role of spontaneous activity, which is present before vision starts, and will no doubt inspire future experimental studies.

**Decision letter after peer review:**

[Editors’ note: the authors submitted for reconsideration following the decision after peer review. What follows is the decision letter after the first round of review.]

Thank you for submitting your work entitled "Adaptation of spontaneous activity in the developing visual cortex" for consideration by *eLife*. Your article has been reviewed by two peer reviewers, and the evaluation has been overseen by a Reviewing Editor and a Senior Editor. The following individual involved in review of your submission has agreed to reveal their identity: József Fiser (Reviewer #3).

Our decision has been reached after consultation between the reviewers. Based on these discussions and the individual reviews below, we regret to inform you that your work will not be considered further for publication in *eLife*.

This manuscript addresses the important and interesting topic of the role of spontaneous activity at different levels of the visual pathway in the emergence of topographically-organized V1 receptive fields. Although the reviewers commended the modelling and data analyses, as well as the quality of the writing, they did not think that this study provides fundamental insights into the biological mechanisms responsible for establishing the connectivity and response properties of cortical neurons. In addition, it was felt that the study lacked novelty in some aspects and that some of the findings presented had been over interpreted.

Reviewer #1:

This paper provides a detailed theoretical analysis of how different types of spontaneous activity influence receptive field development in the cortex. The model is well-grounded, the simulations are supported by nice mathematical analyses, and the paper is well written. Overall I think it's good work and certainly worthy of publication in a solid journal. However I don't think it provides the level of advance appropriate for *eLife*. My main concerns in this regard are as follows.

1) The paper shows how the deleterious effects of H events can be mitigated, but it doesn't explain why H events are present in the first place. Overall the results don't seem very surprising, and I don't think there's a huge amount of new insight into the underlying biological principles.

2) The modelling and analytical tools used have already been thoroughly investigated. For instance Equation 2 is Equation 8.9 in the Dayan and Abbott textbook, the eigenvector analysis is very similar to that in Mackay and Miller, 1990, and many of the principles relevant to this work were already established by Willshaw and von der Malsburg, 1976. While all this certainly supports the soundness of the work presented here, it does undermine its novelty.

3) I think the experimental finding that the amplitude of L and H events preceding a given H event are correlated could be explained simply by long-term fluctations in overall cortical excitability, rather than being a confirmation of the model.

Reviewer #3:

This manuscript aims at reconciling the roles of spontaneous activities of thalamic vs. cortical origin in the establishment of cortical connectivity and known neural characteristics of those activities. The authors propose the existence of an amplitude-adaptation mechanism for the cortically generated spontaneous activity and test this idea by re-analyzing previous in vivo data and perform simulation on a 1D model of cortical development.

I find the paper clearly written, the data analyses and the simulations adequately executed. However, I also think some additional effort to clarify the exact contribution of the paper, the link between analyses and the claims of the paper and its link to previous proposals would be necessary to better assess the significance of the proposed model. In addition, clear predictions justifying the insight of the work would further improve the value of the manuscript.

I had three issues when reading the manuscript, handling of which might help to increase the impact of the paper.

First, it seems to me that describing the exact contribution of the paper should be better elaborated. My understanding is that the course of action of the paper is this: (a) taking up the experimental findings of Siegel et al., 2012, about the two independent sources of activity generation in the developing visual system (thalamic and cortical), the authors tried to fold these constraints to the computational requirements of topographic pattern formation. (b) After choosing one particular implementation, they realized and demonstrated that an adaptive tuning of the global H-events' amplitude is needed for stable behavior in this model. Finally, (c) after reanalyzing the original Siegel et al., 2012, data, they found evidence for such an adaptive mechanism. In addition, they linked their work to the concept of sparsification of neural signal for efficient coding.

If this is correct, I would like to know the answer to the following set of questions.

1) Why the postulation of H-events acting homeostatically? It is clear that this assumption can step in for the missing normalization functionality necessary for the Hebbian plasticity rule to operate properly, but there are other options as well. Did the authors have a more established reason to go with this choice? Beyond being a heterosynaptic learning rule, in what way the resulting Hebbian input-threshold learning rule is different from or more adequate than other realizations (e.g. the ones based on pre-post synaptic activities)? And wrt other heterosynaptic rules?

2) When arguing for evidence in the Siegel et al., 2012, data based on the fact that the amplitude of an H-event and the amplitudes of H- and L-events in the preceding 100 msec are correlated, how can they rule out the alternative that the correlation is not a result of an active adaptation mechanism for H-events, but due to a general fluctuation of both L- and H- event amplitude magnitudes in time caused by other factors? Shouldn't the existence of active adaptation mechanism be supported by showing a causal proportionality based on some aggregate sum of amplitudes and frequencies of preceding events in a specified window rather than just a time-insensitive amplitude-to-amplitude correlation the authors demonstrate?

My second issue is related to the notion of sparsification, which has been widely but typically very loosely used in the literature, and the manuscript seem to follow this trend. For a sufficient treatment of sparsification, see Willmore and Tolhurst, 2001, Berkes et al., 2009 and Zylberberg et al., 2013. A full treatment of sparseness includes (a) proper definitions (i.e. distinguishing between population and lifetime sparseness of neural activity), (b) the clarification that sparsification can be interpreted as a principle either for energy conservation, in which case lifetime sparseness is the appropriate measure, and homeostasis is a possible implementation, or for efficient coding for which population sparseness is the proper measurement, and regulation of the individual firing rates is an insufficient proxy, and (c) using the proper sparseness for the actual argument. The Siegel et al., 2012, paper carefully uses the term "event" sparsification, which refers to the occurrence of waves, bursts, which has only indirect connection to information processing capacity as is was implied (but not clearly spelled out) in Olshausen and Field, 1996. The Rochefort et al., 2009, paper does refer to sparsity of neural activity, but it neglects the fact that by measuring percentage of active neurons in any one event itself does not capture the information processing capacity of the network either.

3) My problem is that it is unclear what the present manuscript aims at: is it simply trying to match the Siegel et al. recorded data with their model (which they show in the manuscript) without any attempt to link that to information processing, or alternatively, they give their model a functional spin by tying to link the results to the concept of efficient coding, in which case a whole different set of analyses is required. According to the authors "…we observe a progressive sparsification of the effective spontaneous events during ongoing development in our model. ". This would suggest that they are describing event statistics. But then in the Discussion, they refer to the efficient coding aspect of spike sparsification for which there is no adequate analysis provided in the manuscript. Given that this is a computational modelling paper, it would help to get a clear statement about and the corresponding analysis supporting the goals the authors have with referring to sparsification. The proposed adaptation of the H-events amplitudes would have a significant effect on efficient coding, and this is a different issue from map formation. I wonder if the authors want to dive into that issue, but in any case, it would be necessary to clarify what type of sparsification the authors discuss in the manuscript.

My third comment is pointing out that the authors did not provide any testable prediction based on their new model, if I am not mistaken.

4) If the new idea is that amplitude-adaptation must exist in the cortex, proposing a test that can verify this directly would be invaluable.

[Editors’ note: further revisions were suggested prior to acceptance, as described below.]

Thank you for submitting your article "Adaptation of spontaneous activity in the developing visual cortex" for consideration by *eLife*. Your article has been reviewed by three peer reviewers, and the evaluation has been overseen by a Reviewing Editor and Andrew King as the Senior Editor. The following individuals involved in review of your submission have agreed to reveal their identity: József Fiser (Reviewer #1); Nicholas V Swindale (Reviewer #2); Jianhua Cang (Reviewer #3).

The reviewers have discussed the reviews with one another and the Reviewing Editor has drafted this decision to help you prepare a revised submission.

Summary:

This modelling study shows that a set of rules involving adaptation can lead to developmental changes in the visual cortex that match those observed experimentally, including receptive field formation, synapse stabilization and a slow sparsification of spontaneous activity. The biological applications of this study are important and interesting. The proposed roles of local low-synchronicity events originating in the retina and global high-synchronicity events originating in the cortex, and the associated predictions from this research, will stimulate future experimental work.

Revisions:

The reviewers were generally positive, but raised several concerns that will need to be addressed satisfactorily before your paper can be considered to be acceptable for publication in *eLife*. While the assessment of this work took into account the revisions made to the previously rejected manuscript, it was considered as a new submission and has therefore attracted additional comments.

1) The direct comparison with the BCM learning rule has increased the value of the paper significantly. Nevertheless, questions were raised about more conventional normalization schemes. We appreciate that it would be excessive to test all existing alternatives. However, alternatives do need to be considered and the authors should explain how these compare to the non-adaptive-augmented version of your original model, and explain why, if only one contrasting alternative is to be tested, BCM is the right choice.

2) The manuscript has benefited from making the main text lighter and shifting much of the standard mathematical treatment to the Appendix. Nevertheless, the reviewers felt that the presentation of the manuscript could be improved further to more clearly explain the proposed links between model behaviour and real development and to make it easier for experimentalists to follow.

3) There was some discussion among the reviewers about the classification of spontaneous events into H and L types. Because this is critical for the study, further explanation of the previous evidence for this would be helpful.

4) The fact that the model itself produces L and H events seemed to throw a spanner into the works. One criticism of the model is that in the real brain, cortical spontaneous activity is not likely to be imposed from outside, as in the model, but is integral to the overall developmental dynamics and might involve circuits beyond the scope of the model that could regulate plasticity if needed. But now the significance of the imposed events seems to be called into question. It certainly does not help that H events can now be effectively identified as L events, resulting in confusion as to how to interpret what was going on and hence to evaluate the significance of the model.

5) Figure 2: the initial conditions were noisy but arguably not much different in other respects from the final state. Can you use much broader and more scattered initial receptive field widths? Is more than a bit of smoothing and weight thresholding going on?

6) The method used to show adaptation in the experimental data is unclear. Surely the exponential is used to scale the weights of the values that go into the running average, not their amplitude? Assuming that is the case, what is the point of having a time constant for the averaging of 1000s when recordings do not last longer than 300 s? Effectively it is an unweighted average, or close to it.

7) Can the correlation between the amplitude of an H event and the average amplitude of preceding events shown in Figure 5B be more parsimoniously explained by slow changes in the overall signal strength over time? This was felt to be a key issue.

---

## [Author Response]

[Editors’ note: the authors resubmitted a revised version of the paper for consideration. What follows is the authors’ response to the first round of review.]

Reviewer #1:This paper provides a detailed theoretical analysis of how different types of spontaneous activity influence receptive field development in the cortex. The model is well-grounded, the simulations are supported by nice mathematical analyses, and the paper is well written. Overall I think it's good work and certainly worthy of publication in a solid journal. However I don't think it provides the level of advance appropriate for eLife. My main concerns in this regard are as follows.

We thank the reviewer for the critical assessment of our work. However, we were puzzled by the reviewer’s evaluation that our work does not provide the appropriate advance for *eLife* after evaluating it as good and worthy of publication in a solid journal. Our work provides a balance between a mathematically-grounded theoretical framework and modeling applied to published experimental data, with some predictions already supported by the data and further predictions to be tested experimentally. In our view, this integrated approach is precisely what a journal like *eLife* aims to publish.

1) The paper shows how the deleterious effects of H events can be mitigated, but it doesn't explain why H events are present in the first place. Overall the results don't seem very surprising, and I don't think there's a huge amount of new insight into the underlying biological principles.

We acknowledge that our discussion about the biological principles underlying H-events and their potential role in developmental synaptic refinements were unclear.

Even though global spontaneous cortical events (like the H-events) have been observed in different sensory cortices during development (e.g., auditory cortex (Babola et al., 2018), also visual cortex (Gribizis et al., 2019)), their functional role remains elusive. We addressed this knowledge gap in our modeling with the hypothesis that they are a self-regulating homeostatic mechanism operating in parallel to network refinements driven by peripheral spontaneous activity (L-events). Although we note, that the goal of our work is not to propose a mechanism for the generation of H-events (which is a complex process involving inhibition and multiple cell types, see for e.g. (Leighton et al., 2020)). We propose three key reasons for the hypothesis about the homeostatic role of H-events, which we have included in the revised manuscript:

a) Since all cells are maximally active during H-events, these patterns likely do not carry much information that can be used for synaptic refinement.

b) Synaptic scaling, a well-known homeostatic mechanism, has been shown to induce global synaptic depression in response to highly-correlated network activity (like H-events) (Turrigiano and Nelson, 2004).

c) As postulated in the synaptic homeostasis hypothesis (SHY), during slow-wave sleep, which is characterized by highly synchronous activity (like H-events), synaptic strengths are downscaled to balance their net increase during wakefulness (Tononi and Cirelli, 2006). Since during development sleep patterns are not yet regular, we reasoned that refinement (by L-events) and homeostasis (by H-events) occur simultaneously instead of being separated into wake and sleep states.

As we further discuss in response to point 2 below, the proposed adaptation in H-event amplitudes has not been previously studied. Importantly, going beyond the resulting mitigation of deleterious effects caused by too strong H-events, this adaptation surprisingly also explains the developmental sparsification of spontaneous activity on a much longer timescale, without any additional finetuning.

Additionally, we followed reviewer 3’s suggestion to consider alternative learning rules to the Hebbian rule (specifically, the Bienenstock, Cooper and Munro, or BCM, rule) (Bienenstock et al., 1982). We found that in the BCM rule the functional role of H-events is mechanistically different than in the Hebbian rule. In particular, H-events are not deleterious and do not cause cortical decoupling as in the Hebbian rule. Instead, they regulate the threshold between potentiation and depression hence act homeostatically like we originally postulated. We now include an extended set of results on the BCM rule in the revised manuscript, and compare the role of H-events in synaptic refinements when using both the BCM and Hebbian rules (updated Figure 3 and corresponding text).

2) The modelling and analytical tools used have already been thoroughly investigated. For instance Equation 2 is Equation 8.9 in the Dayan and Abbott textbook, the eigenvector analysis is very similar to that in Mackay and Miller, 1990, and many of the principles relevant to this work were already established by Willshaw and von der Malsburg, 1976. While all this certainly supports the soundness of the work presented here, it does undermine its novelty.

We respectfully disagree with the reviewer on this point. The Hebbian learning rule that we study is fundamental to studying synaptic plasticity. In fact, different forms of spike-timingbased STDP based on spikes and triplets of spikes can be reduced to the same type of equation (Kempter et al., 1999, Gjorgjieva et al., 2011). In our view, applying a fundamental theoretical framework to an interesting biological question does not undermine the novelty of our paper. For comparison, these rules have been used by numerous studies in different settings (see also a recent review by Zenke et al. (2017), Equation 2.2), including some published in *eLife*:

• (Sweeney and Clopath 2020) (*eLife* 2020) – Their Equation 2 for the study of population coupling and plasticity of cortical responses

• (Weber and Sprekeler 2018) (*eLife* 2018) – Their Equation 2 for the learning of place and grid cells

**•** (Bono and Clopath 2018) (PLoS Comp Biol 2019)– Their Equation 3 for the study of ocular dominance plasticity

**•** (Toyoizumi et al. 2013) (Neuron 2013) – Equation on line 18, pg. 54 (Equation is unlabeled) for the study of critical period plasticity

We used this theoretical framework to implement Hebbian plasticity and straightforwardly test the hypothesis of the homeostatic role of H-events (see point 1). Importantly, we derived that this framework with fixed properties of L- and H-events cannot generate stable and refined receptive fields, and hence proposed the adaptation of H-event amplitude. This advances the field by combining plasticity based on fixed activity statistics operating on a shorter timescale of a single postnatal day, and the long-term developmental modification of activity statistics on the timescale of several days. Such combination of timescales is unique to the developmental setting of the biological problem that we consider and has not been previously addressed. At the same time, our work goes beyond the topographic map formation by correlated activity proposed by (Willshaw and von der Malsburg, 1976) in that it focuses on the generation of stable and robust receptive fields, using in vivo spontaneous activity patterns, which could not be recorded in 1976. We show that this refinement of receptive fields leads to the developmental sparsification of spontaneous events. To our knowledge, no previous work has demonstrated this feedback from connectivity to activity on a developmental timescale. We now highlight these advances in the revised manuscript and make several experimental predictions in the Discussion.

In response to the reviewer’s comment, we moved the mathematical analysis to an Appendix (Appendix 1: Figures 1–4 and corresponding text) in the revised manuscript because it is not the main result of our work. Given that our paper is not a mathematical paper about a new technique, we do not believe that the use of well-established methods should preclude publication in *eLife*. Moreover, the eigenvector analysis is a standard mathematical technique from dynamical systems theory to study the phase plane of the synaptic weights and has been used by multiple theoretical studies before us (Litwin-Kumar and Doiron, 2014, Gjorgjieva et al., 2011, Toyoizumi et al., 2014, among others). Including the full eigenvector analysis (Appendix 1: Figure 2) is necessary to establish a relationship between the abstract parameters in the model and in vivo measured spontaneous activity properties. Considering the broad readership of *eLife*, we believe that it is important to present the analysis as part of the Appendix in our revised manuscript to make it as clear and self-contained as possible. Even so, we extended the mathematical analysis of MacKay and Miller, 1990, in several novel ways:

a) the derivation of the entries of the input correlation matrix for L-events (Equation 17-19),

b) the calculation of the receptive field size (Equation 21-27, and comparison to simulations in Appendix 1: Figure 1),

c) the reduction into a 2D phase plane for the weights corresponding (or complementary) to the receptive field (Figure 4D and Appendix 1: Figures 3–4),

d) the analysis for how adaptive H-events successfully refine receptive fields without cortical decoupling (Figure 4E).

3) I think the experimental finding that the amplitude of L and H events preceding a given H event are correlated could be explained simply by long-term fluctations in overall cortical excitability, rather than being a confirmation of the model.

We thank the reviewer for raising this important point. To explore this possibility, we performed two additional analyses of our data.

First, we analyzed consecutive recordings (each ∼5 mins long) in the same animal of which we had between 3 and 14 in all 26 animals to identify possible fluctuations on a longer timescale. We found that the average amplitude of all (L and H) events is not significantly different across consecutive recordings of the same animal (Figure 5—figure supplement 1A in the revised manuscript, one-way ANOVA tests, *p >* 0.05 in 23 out of 26 animals). Across different animals and ages, individual event amplitudes remained uncorrelated between successive recordings at this timescale, which we confirmed by plotting the difference in event amplitude as a function of the time between recordings (Figure 5—figure supplement 1B in the revised manuscript). Hence, our additional analysis shows that long-term fluctuations in cortical excitability are unlikely to account for the observed correlation.

Next, even though we did not find evidence for long-term fluctuations in the overall cortical excitability in the data, we used our model to investigate whether such fluctuations could be an alternative mechanism to adaptation for the stable and robust refinement of receptive fields under the Hebbian rule with L- and H-events. To simulate a top-down input that generates longterm fluctuations in L- and H-events, we sampled the event amplitudes from correlated OrnsteinUhlenbeck processes, according to:Lamp(t)=[αL+pX1(t)=X2(t)2+1−pX1(t)]+Hamp(t)=[αH+pX1(t)=X2(t)2+1−pX2(t)]+where *α_L_*and *α_H_*are constants, *ρ* is the correlation and *X*_1_(*t*)*,Χ2*(*t*) are sampled as (Gillespie, 1996):X(t+Δt)=X(T)μ+c(1−μ2)2n, where *µ* = *e*^−∆*t/τ*^, *τ* is the relaxation time of the process, *c* is the diffusion constant, and *n* is a random number drawn from a normal distribution of mean 0 and standard deviation 1. For all the simulations, we fixed *α_L_*= 1, *α_H_*= 6, *τ* = 1 s, *c* = 0.1 and ∆*_t_* = 0.01 s, and explored correlation strengths of *ρ* = 0, 0.5 and 1. During L- (H-events), the amplitudes of all participating cells will be *L*_amp_(*t*) (*H*_amp_(*t*)). Apart from the time-dependent amplitudes, we followed the same protocol of simulations from our manuscript and quantified the receptive field size (*S*) for different levels of correlation between L- and H- amplitudes (Figure 5—figure supplement 2A). We also quantified the proportion of simulations that resulted in selective, non-selective and decoupled receptive fields (Figure 5—figure supplement 2A, compare to Figure 3C in the revised manuscript). For the three values of correlation strengths that we tested, the proportion of simulations that resulted in selective receptive fields is always smaller than 20%, which is in the same range as that for non-adaptive H-events (Figure 5—figure supplement 2A, compare to Figure 3C left in the revised manuscript). If the reviewer wants us to include these results in the revised manuscript, we would be happy to do so.

As an alternative to these slowly-varying event amplitudes, we also considered whether a topdown signal that decays on a slow developmental timescale that affects only H-event amplitudes without directly integrating ongoing activity in the network might generate selectivity and prevent cortical decoupling. In this set of simulations, H-events are initially very strong but their amplitudes decay exponentially as a function of developmental time with different time constants (Author response image 1). To determine the decay time constants used in this model, we fitted an exponential decay function to the initial decay of amplitude of the model with adaptive H-events (Author response image 1). The only working solution in this scenario was to consider very fast decays, when H-events are effective only at the beginning of the simulation, which reduces to the case of pure L-events in our manuscript (Author response image 1). We do not consider this a biologically plausible scenario since H-events are detected in animals as old as P14 (Figure 3A of Siegel et al., 2012). If the reviewer wants us to include these results in the manuscript, we would be happy to do so.

**Author response image 1. respfig1:** A top-down signal imposes exponential decay of H-event amplitudes during development (independent of ongoing network activity) enhances receptive field refinement only if the decay is very fast. A. Average amplitudes of adaptive H-events as a function of simulation time for two input thresholds in the Hebbian learning rule, θu = 0.6 and θu = 0.65. To determine the decay time constants used in B and C, an exponential decay function was fitted to the initial decay of amplitude, with fitted decay time constants τH = 825 s and τH = 676 s, respectively. B. Using the range of time constants from A, we simulated the model with non-adaptive H-events. Their amplitudes decayed exponentially with different decay time constants (τH = {10,100,1000,10000} s). The inset shows a detail of the beginning of the simulation. C. Final receptive fields for non-adaptive H-events with exponentially decaying amplitudes as in B for the two different input thresholds.

Reviewer #3:This manuscript aims at reconciling the roles of spontaneous activities of thalamic vs. cortical origin in the establishment of cortical connectivity and known neural characteristics of those activities. The authors propose the existence of an amplitude-adaptation mechanism for the cortically generated spontaneous activity and test this idea by re-analyzing previous in vivo data and perform simulation on a 1D model of cortical development.I find the paper clearly written, the data analyses and the simulations adequately executed. However, I also think some additional effort to clarify the exact contribution of the paper, the link between analyses and the claims of the paper and its link to previous proposals would be necessary to better assess the significance of the proposed model. In addition, clear predictions justifying the insight of the work would further improve the value of the manuscript.I had three issues when reading the manuscript, handling of which might help to increase the impact of the paper.First, it seems to me that describing the exact contribution of the paper should be better elaborated. My understanding is that the course of action of the paper is this: (a) taking up the experimental findings of Siegel et al., 2012, about the two independent sources of activity generation in the developing visual system (thalamic and cortical), the authors tried to fold these constraints to the computational requirements of topographic pattern formation. (b) After choosing one particular implementation, they realized and demonstrated that an adaptive tuning of the global H-events' amplitude is needed for stable behavior in this model. Finally, (c) after reanalyzing the original Siegel et al., 2012 data, they found evidence for such an adaptive mechanism. In addition, they linked their work to the concept of sparsification of neural signal for efficient coding.

Indeed, the reviewer is correct, this is a beautiful way of summarizing our work (apart from the link to the concept of sparsification for efficient coding that we addressed). We have revised the manuscript to better highlight its main contribution following many of the reviewer’s suggestions.

If this is correct, I would like to know the answer to the following set of questions.1) Why the postulation of H-events acting homeostatically? It is clear that this assumption can step in for the missing normalization functionality necessary for the Hebbian plasticity rule to operate properly, but there are other options as well. Did the authors have a more established reason to go with this choice? Beyond being a heterosynaptic learning rule, in what way the resulting Hebbian input-threshold learning rule is different from or more adequate than other realizations (e.g. the ones based on pre-post synaptic activities)? And wrt other heterosynaptic rules?

We agree with the reviewer’s feedback that we did not motivate this assumption, which inspired us to study the Hebbian learning rule in the first place. We propose three key reasons for this motivation, which we have included in the revised manuscript:

a) Since all cells are maximally active during H-events, these patterns likely do not carry much information that can be used for synaptic refinement.

b) Synaptic scaling, a well-known homeostatic mechanism, has been shown to induce global synaptic depression in response to highly-correlated network activity (like H-events) (Turrigiano and Nelson, 2004).

c) As postulated in the synaptic homeostasis hypothesis (SHY), during slow-wave sleep, which is characterized by highly synchronous activity (like H-events), synaptic strengths are downscaled to balance their net increase during wakefulness (Tononi and Cirelli, 2006). Since during development sleep patterns are not yet regular, we reasoned that refinement (by L-events) and homeostasis (by H-events) occur simultaneously instead of being separated into wake and sleep states.

Based on the reviewer’s comment, we implemented an alternative learning rule that has had a prominent application in previous theoretical models of synaptic plasticity, specifically, the Bienenstock, Cooper and Munro, or BCM, rule (Bienenstock et al., 1982). Synaptic potentiation and depression in the BCM rule are still determined by Hebbian terms that require coincident pre- and postsynaptic activity, but with two important differences to the Hebbian rule: (1) the rule depends on the square of postsynaptic activity (in contrast to Hebb’s which depends linearly on postsynaptic activity), and (2) the rule has the property of metaplasticity where the threshold which determines the amount of potentiation vs. depression is modulated as a function of postsynaptic activity. We hypothesized that this adaptive way to regulate potentiation vs. depression might be sufficient to replace the requirement for our proposed adaptive H-events in the Hebbian rule. Both Hebbian and BCM rules are rate-based learning rules and have been extensively analyzed in previous works (e.g. Cooper and Bear (2012) for a review). Importantly, however, they can be directly linked to different forms of spike-timing-dependent plasticity (STDP) which depend on the order and timing of pairs and triplets of spikes, respectively, and hence have even wider applicability. For instance, the pair-based STDP rule maps to the Hebbian rule (Kempter et al., 1999) while the triplet STDP rule maps to the BCM rule under certain conditions of Poisson spiking (Pfister and Gerstner, 2006, Gjorgjieva et al., 2011).

We found that the BCM learning rule generates refined receptive fields without cortical cell decoupling from the thalamus (Figures 3B, C in the revised manuscript). Indeed, under the BCM rule, H-events are homeostatic by dynamically sliding the threshold between potentiation and depression. However, when analyzing the receptive field properties, we found that the BCM rule generates receptive fields with much worse topography than the Hebbian rule (Figure 3D in the revised manuscript). The underlying reason for the worse topography is the fact that small Levents, which have precise information for topographic connectivity refinements, mostly cause LTD in the synaptic weights (Figure 3E, F). Indeed, the sliding threshold that determines the amount of potentiation vs. depression is systematically increased both by large L-events and H-events. Therefore, the cortical activity triggered by small L-events is often lower than the sliding threshold at the event onset, resulting in LTD. We now provide an extensive discussion of the results with the BCM rule and specifically compare them to the Hebbian rule with adaptive H-events in the revised manuscript.

2) When arguing for evidence in the Siegel et al., 2012, data based on the fact that the amplitude of an H-event and the amplitudes of H- and L-events in the preceding 100 msec are correlated, how can they rule out the alternative that the correlation is not a result of an active adaptation mechanism for H-events, but due to a general fluctuation of both L- and H- event amplitude magnitudes in time caused by other factors? Shouldn't the existence of active adaptation mechanism be supported by showing a causal proportionality based on some aggregate sum of amplitudes and frequencies of preceding events in a specified window rather than just a time-insensitive amplitude-to-amplitude correlation the authors demonstrate?

In our answer to P1.3, we addressed the point of existing fluctuations in the data by a further analysis where we compared the average amplitude of L- and H-events across recordings in the same animal, and determined that long-term fluctuations in cortical excitability are unlikely to account for the observed correlation (Figure 5—figure supplement 1 in the revised manuscript). We also performed additional simulations where we used such hypothetical correlated fluctuations between L- and H-events as input, as well as assumed the decay of H-event amplitude uncoupled to ongoing activity, and showed that neither can generate robust and refined receptive field refinement.

We followed the reviewer’s suggestion to perform a time-sensitive analysis of the correlation between H-events amplitude and average preceding activity by re-analyzing the data with a time-dependent leak in the amplitudes of events preceding each H-event. The amplitude of each spontaneous event registered ∆*t_i_*seconds before an H-event was multiplied by an exponential kernel exp(−∆*t_i_/τ*), and then all amplitudes were averaged. Due to the leak, spontaneous events that are closer in time to the H-event contribute more to the average, which mimics the mechanism used in the implementation of H-event adaptation in the simulations. For this analysis, we concatenated several consecutive 5 min-long recordings in the data to have longer time windows for the aggregate analysis. We investigated a range of time constants *τ* and found that as long as this time constant is not too fast (i.e. only a single recent event influences the amplitude of a new H-event), the correlation between the amplitude of H-events and the average amplitude of preceding activity with the leak remains significant. We present this analysis in Figure 5 and its supplement in the revised manuscript.

My second issue is related to the notion of sparsification, which has been widely but typically very loosely used in the literature, and the manuscript seem to follow this trend. For a sufficient treatment of sparsification, see Willmore and Tolhurst, , 2001, Berkes et al., 2009 and Zylberberg et al., 2013. A full treatment of sparseness includes (a) proper definitions (i.e. distinguishing between population and lifetime sparseness of neural activity), (b) the clarification that sparsification can be interpreted as a principle either for energy conservation, in which case lifetime sparseness is the appropriate measure, and homeostasis is a possible implementation, or for efficient coding for which population sparseness is the proper measurement, and regulation of the individual firing rates is an insufficient proxy, and (c) using the proper sparseness for the actual argument. The Siegel et al., 2012, paper carefully uses the term "event" sparsification, which refers to the occurrence of waves, bursts, which has only indirect connection to information processing capacity as is was implied (but not clearly spelled out) in Olshausen and Field, 1996. The Rochefort et al., 2009, paper does refer to sparsity of neural activity, but it neglects the fact that by measuring percentage of active neurons in any one event itself does not capture the information processing capacity of the network either.

We thank the reviewer for this important feedback on how we discuss the sparsification of spontaneous activity and the literature suggestions. We completely agree that our definition of sparsification was unclear and incomplete. Our notion of sparsification follows that of Siegel et al., 2012, and it refers to an overall sparsification of network events (fewer active cells per event) where cortically-generated H-events become replaced with peripherally-driven L-events. Hence, it has only indirect connection to information processing capacity of the network which we incorrectly alluded to in our manuscript. We have clarified this in our revised manuscript both in the Results and the Discussion sections, and in the text we now refer to it as “event sparsification”. We also provide relevant citations for alternative notions of sparsification.

3) My third comment is pointing out that the authors did not provide any testable prediction based on their new model, if I am not mistaken.4) If the new idea is that amplitude-adaptation must exist in the cortex, proposing a test that can verify this directly would be invaluable.

We agree with the reviewer’s suggestion to highlight testable predictions based on our modeling results, which were not clearly stated in our previous manuscript.

We included a new section in the Results (“Modulating spontaneous activity properties makes different predictions for receptive field refinements”) and also in the Discussion (“Predictions of the model”) where we elaborate these model predictions and propose experimental validations.

1) First, we predict that under the Hebbian learning rule, changing the frequency of adaptive Hevents can affect the size of the resulting receptive fields (Figure 4 in the revised manuscript). The frequency of H-events can be experimentally manipulated, for instance, by a gap junction blocker (carbenoxolone) that reduces the frequency of H-events (Siegel et al., 2012). According to our prediction, cortical receptive fields will be broader after this manipulation.

2) Next, we predict receptive field size under the Hebbian learning rule when varying the size of L-events (Figure 6 in the revised manuscript). Specifically, larger L-events will generate larger receptive field sizes and worse topography. There are at least two ways to manipulate L-events experimentally.

(2.1) One way is to manipulate the source of L-events in the cortex, namely the retinal waves. Our predictions can be related to previous experimental studies that manipulated the size of retinal waves in the *β*2 knockout mouse and found less refined receptive fields in the cortex (Sun et al., 2008, Stafford et al., 2009, Cutts and Eglen, 2014).

(2.2) Another way is to directly manipulate L-events in the cortex. Recently it has been shown that the size of L-events in the developing visual cortex increases upon suppression of somatostatin-positive interneurons (Leighton et al., 2020). After such a manipulation, our work predicts larger receptive fields and worse topography.

3) Third, our model generates “event sparsification,” with H-events being gradually replaced by L-events during development. This can be tested by examining L- and H-events properties in older animals (e.g., P14) and comparing them to the properties in younger animals (e.g., P8-P10). While other experimental studies report such developmental event sparsification (Rochefort et al., 2009, Frye and MacLean, 2016, Smith et al. 2015, Ikezoe et al., 2012, Shen and Colonnese, 2016, Golshani et al., 2009), in many of these studies activity has not been segregated into peripherally driven L-events and cortically generated H-events. Therefore, our model predicts that the frequency of L-events would increase, while the frequency of H-events would decrease over development.

4) Finally, we propose that for a Hebbian rule to drive developmental refinements of receptive fields, the amplitude of H-events should adapt to the ongoing network activity. The confirmation of such a fast adaptation mechanism would require prolonged and detailed activity recordings in vivo, which are within reach of modern technology (Ackman and Crair, 2014, Gribizis et al., 2019). Our work also predicts that manipulations that affect overall activity levels of the network, such as activity reduction by eye enucleation, would correspondingly affect the amplitude of ongoing H-events.

[Editors’ note: what follows is the authors’ response to the second round of review.]

Revisions:The reviewers were generally positive, but raised several concerns that will need to be addressed satisfactorily before your paper can be considered to be acceptable for publication in eLife. While the assessment of this work took into account the revisions made to the previously rejected manuscript, it was considered as a new submission and has therefore attracted additional comments.1) The direct comparison with the BCM learning rule has increased the value of the paper significantly. Nevertheless, questions were raised about more conventional normalization schemes. We appreciate that it would be excessive to test all existing alternatives. However, alternatives do need to be considered and the authors should explain how these compare to the non-adaptive-augmented version of your original model, and explain why, if only one contrasting alternative is to be tested, BCM is the right choice.

We are happy that the reviewers value the new comparison with the BCM learning rule. We also appreciate the questions regarding alternative, more conventional normalization schemes for the original Hebbian covariance learning rule without our proposed adaptation of cortical H-events. A common approach to prevent unconstrained weight growth in previous theoretical work is to limit the total weight strength for a cell. Two methods of enforcing such a constraint have been proposed: subtractive and multiplicative normalization (Miller and Mackay, 1994).

Subtractive normalization constrains the sum of all weights to a constant value by subtracting from each weight a constant amount that depends on the sum of all weights. We now demonstrate that the input threshold in the learning rule, which determines whether weights are potentiated or depressed, and the inclusion of H-events, implement subtractive normalization (see Appendix section “Normalization constraints”). Previous theoretical work has shown that subtractive normalization induces strong weight competition and drives the emergence of weight selectivity and refined receptive fields (Miller and Mackay, 1994). However, our results in Figure 3 demonstrate that this normalization constraint is insufficient to stabilize the receptive fields (i.e., prevent the weights from completely depressing) due to the stronger cortical H-events relative to the peripheral L-events.

Multiplicative normalization constrains the sum of all weights to a constant value by subtracting from each weight an amount that is proportional of the weight itself. We did not consider a multiplicative normalization constraint because previous work has already shown that such a constraint does not generate refined (sharpened) receptive fields (see Miller and Mackay, 1994, Dayan and Abbott, 2001). Rather, multiplicative normalization yields a graded receptive field where most mutually correlated inputs are represented (Miller and Mackay, 1994).

A third alternative to introduce weight competition (and hence emergence of selectivity and refined receptive fields) is the BCM rule. We contrasted the BCM rule with the Hebbian covariance learning rule in greatest detail for the following reasons (see text between Equations 2 and 3):

i) The BCM rule has been shown to generate selectivity in postsynaptic neurons that experience patterned inputs. Specifically, the framework can explain the emergence of ocular dominance (neurons in V1 being selective for input from one of the two eyes) and orientation selectivity in the visual system (Cooper et al., 2004).

ii) It implements homeostatic weight regulation by “sensing” ongoing activity in the network and adjusting weights in the network to maintain a target level of ongoing activity (as opposed to constraining the sum of the weights in the cases above). Hence, it is more comparable to our adaptive mechanism that “senses” ongoing activity and adjusts H-event amplitude. The difference is that in the BCM rule weights are adjusted, while in our model, cortically-generated activity is adjusted.

We now also include a discussion of these alternative normalization schemes and explain why we chose to directly contrast our results to those of the BCM rule (see Results section “A network model for connectivity refinements driven by spontaneous activity”, and third paragraph of the Discussion section “Assumptions in the model”).

2) The manuscript has benefited from making the main text lighter and shifting much of the standard mathematical treatment to the Appendix. Nevertheless, the reviewers felt that the presentation of the manuscript could be improved further to more clearly explain the proposed links between model behaviour and real development and to make it easier for experimentalists to follow.

We greatly value this feedback from the reviewers. Therefore, we took additional effort to explain the links between model behavior and real development throughout the manuscript. This is most prominent in our Introduction and Discussion, but also in the Results section “Adaptive H-events promote the developmental event sparsification of cortical activity.”

3) There was some discussion among the reviewers about the classification of spontaneous events into H and L types. Because this is critical for the study, further explanation of the previous evidence for this would be helpful.

Thank you – this is a great point. We used the classification of events into L and H based on Siegel et al., 2012. To classify spontaneous events observed in the visual cortex, Siegel et al., 2012, used a clustering analysis based on two features: the average amplitude and jitter (a measure related to synchrony) of events recorded in vivo using two-photon calcium imaging. This analysis revealed that events with 20-80% participation rate are statistically different from events with 80-100% participation rate (see their Figure 3D). Accordingly, 20-80% participation rate events have lower amplitudes and low synchronicity (hence called L-events), while 80-100% participation rate events have higher amplitudes and high synchronicity (hence called H-events). Based on additional experiments where the eyes were enucleated or retinal waves enhanced, Siegel et al., 2012, also reported that these events have different sources: retinal waves in the case of L-events and intra-cortical activity in the case of H-events.

The classification of events according to the participation rate of the cells in an event was also recently revisited in Leighton et al., 2020 based on new experiments in the Lohmann lab. Pairing two-photon calcium imaging of L2/3 with simultaneous whole-cell recordings in vivo in V1 neurons during development, the authors found that neurons participating in H-events fire significantly more action potentials than in L-events. The duration of spiking was also significantly longer during H- than L-events. An independent hierarchical clustering using the number of spikes and the duration of spiking confirmed an optimum of two event clusters with a split similar to the one reported in Siegel et al., 2012. A complementary analysis at the population level in wide-field recordings confirmed the existence of two types of events also at this larger scale, despite the different mean calcium amplitudes and event sizes. We now detail in the text how the original classification of events was implemented in Siegel et al., 2012 and the recent validation by Leighton et al., 2020 (second paragraph in the Results section “A network model for connectivity refinements driven by spontaneous activity”).

Nonetheless, for our model, the exact point of division into L- and H-events is not too important. What matters is that we have (1) peripheral events originating in the first input layer of the network (corresponding to the thalamus), which activate a smaller fraction of the neurons, and (2) cortical events originating in the second output layer of the network (corresponding to the primary visual cortex), which activate a much larger fraction of the neurons. Indeed, our results persist when we repeat the analysis with Monte Carlo simulations with L- and H-events being separated by a 70%, as opposed to 80% participation rate threshold. Then L-events activate 20–70% of the thalamic neurons and H-events activate 70–100% of cortical neurons (Author response image 2). All other parameters are the same as in Table 1 of the manuscript. The proportion of simulation outcomes classified as “selective” is comparable to the results presented in the main text (Figure 3C for the Hebbian covariance rule with non-adaptive H-events and Figure 4B for the Hebbian covariance rule with adaptive H-events).

We also explain this in the text (first paragraph of Results section “Spontaneous cortical H-events disrupt topographic connectivity refinement in the Hebbian covariance and BCM plasticity rules” and second paragraph of Results section “Adaptive H-events achieve robust selectivity”).

**Author response image 2. respfig2:** Receptive field statistics for a different distribution of L- and H-event sizes (20–70% and 70–100% participation rates, respectively). A. 300 Monte Carlo simulations with non-adaptive H-events: receptive field size (left) and proportion of simulations that resulted in selective, non-selective and decoupled receptive fields (right). Compare to Figure 3C. B. Same as A for adaptive H-events in the same range of parameters. Compare to Figure 4B.

4) The fact that the model itself produces L and H events seemed to throw a spanner into the works. One criticism of the model is that in the real brain, cortical spontaneous activity is not likely to be imposed from outside, as in the model, but is integral to the overall developmental dynamics and might involve circuits beyond the scope of the model that could regulate plasticity if needed. But now the significance of the imposed events seems to be called into question. It certainly does not help that H events can now be effectively identified as L events, resulting in confusion as to how to interpret what was going on and hence to evaluate the significance of the model.

We thank the reviewers for pointing out these critical issues. Indeed, the model does not include a mechanism for the generation of L- and H-events, but L-events in the input layer and H-events in the output layer of our network are always imposed as external inputs. The reason for this gross simplification is our aim to focus solely on the plasticity of synaptic connections between the sensory periphery (where L-events originate) and the cortex (where H-events originate) induced by these activity patterns (as we note in the Discussion “Assumptions in the model”). We completely agree that a more complete network model of the developing cortex should generate its own spontaneous activity through developmental dynamics and additional circuit elements. In fact, how activity shapes the plasticity of synaptic connections (the question that we address) and how in turn, the change of connectivity affects the generation of activity is a fascinating question that has inspired a few of our recent studies (Montangie et al., 2020, Wu et al., 2020), also in the context of cortical development (Kirchner et al., 2020). However, although recent studies are beginning to unravel the factors involved in the generation of L- and H-events (including inhibition and the role of different interneurons subtypes, as well as the structure of recurrent connectivity), see for example Leighton et al., 2020, the mechanistic underpinnings are unknown and provide us with inspiration for future work.

We now realize that the re-classification of spontaneous events into L and H in the cortical layer of our model (Figure 7) is confusing. Since this re-classification is not necessary for the conclusions of our analysis in Figure 7, we removed it from the text. We now make the point that we do not know if the same criteria based on event participation rates and amplitude can be used to separate the spontaneous events into L and H at later developmental ages (first two paragraphs in Results section “Adaptive H-events promote the developmental event sparsification of cortical activity”). Therefore, we analyzed all spontaneous events of simulated cortical neurons during the process of receptive field refinement in the presence of adaptive H-events (Figure 7B, highlighted in gray). Due to the progressive receptive field refinements and the continued H-event adaptation in response to resulting activity changes, spontaneous events in our model progressively sparsify during ongoing development, meaning that spontaneous events become smaller in size with fewer participating cells.

We modified Figure 7 and the Results section “Adaptive H-events promote the developmental event sparsification of cortical activity” accordingly.

5) Figure 2: the initial conditions were noisy but arguably not much different in other respects from the final state. Can you use much broader and more scattered initial receptive field widths? Is more than a bit of smoothing and weight thresholding going on?

We thank the reviewers for this observation. In fact, by mistake we used a different colormap axis for the initial and final conditions in Figure 2C, hence giving the wrong impression that the initial bias was too strong. We now used the correct colormap in Figure 2 in the manuscript.

Additionally, we ran further simulations where we varied the spread and strength of the initial topographical bias (Figure 3—figure supplement 1). Unless the bias is very broad and weak, topographically organized receptive fields still form.

6) The method used to show adaptation in the experimental data is unclear. Surely the exponential is used to scale the weights of the values that go into the running average, not their amplitude? Assuming that is the case, what is the point of having a time constant for the averaging of 1000s when recordings do not last longer than 300 s? Effectively it is an unweighted average, or close to it.

The reviewers are correct; the exponential kernel was used to scale the amplitude of the events that are included in the running average.

We agree that the second point is confusing because we did not clearly explain our averaging procedure. Indeed, each individual recording in a given animal is ~5 mins (300 s) long. For many of the animals, multiple such (mostly) consecutive recordings were obtained (see for e.g., Figure 5—figure supplement 1A). Therefore, we chose to concatenate such consecutive recordings, usually giving us 40 mins and sometimes even up to 70 mins of continuous data. When computing the running average of amplitudes for all events preceding a given H-event, we only looked Tmax seconds before that H-event. In this time window, we then scaled the amplitude of the events with an exponentially decaying kernel with a time constant of τdecay. In Figure 5B in the manuscript, we showed the correlation for a single choice of these parameters: Tmax = 300 seconds and τdecay = 1000 seconds. In Figure 5—figure supplement 2C and D, we explored the correlation for other values of one of these parameters while fixing the other parameter.

We added more details about our data analysis to the Results section “in vivo spontaneous cortical activity shows a signature of adaptation”.

7) Can the correlation between the amplitude of an H event and the average amplitude of preceding events shown in Figure 5B be more parsimoniously explained by slow changes in the overall signal strength over time? This was felt to be a key issue.

We thank the reviewers for raising this important point. In our manuscript, we included two supplementary figures that demonstrate the absence of such slow fluctuations in the recordings.

First, we analyzed consecutive recordings (each ∼5 mins long) in the same animal of which we had between 3 and 14 in all 26 animals to identify possible fluctuations on a longer timescale. We found that the average amplitude of all (L and H) events is not significantly different across consecutive recordings of the same animal (Figure 5—figure supplement 1A in the manuscript, one-way ANOVA tests, p > 0.05 in 23 out of 26 animals). Second, we found that across different animals and ages, individual event amplitudes remained uncorrelated between successive recordings at this timescale. We confirmed this by plotting the difference in event amplitude as a function of the time between recordings (Figure 5—figure supplement 1B in the manuscript). Hence, our data analysis shows that slow fluctuations in cortical excitability are unlikely to account for the observed correlation. We now include a discussion of this analysis in greater detail in the text (first paragraph in the Results section “in vivo spontaneous cortical activity shows a signature of adaptation”).

Although we did not find evidence for slow fluctuations in the overall data, we used our model to investigate whether such fluctuations could be an alternative mechanism to adaptation of H-event amplitude for the stable and robust refinement of receptive fields. We studied two different scenarios:

1) We first simulated a top-down input (of an unknown source) that generates slow correlated fluctuations in L- and H-events. To achieve this, we sampled amplitudes of L- and H-events from correlated Ornstein-Uhlenbeck processes, according to:Lamp(t)=[αL+ρX1(t)+X2(t)2+1−ρX1(t)]+Hamp(t)=[αH+ρX1(t)+X2(t)2+1−ρX2(t)]+where αL and αH are constants, ρ is the correlation and X1(t), X2(t) are sampled as (Gillespie, 1996):X(t+Δt)=X(t)μ+c(1−μ2)2n where μ=e−Δtτ, τ is the relaxation time of the process, c is the diffusion constant, and n is a random number drawn from a normal distribution of mean 0 and standard deviation 1. We fixed αL=1, αH=6, τ=1 s, c=0.1 and Δt=0.01 s, and explored correlation strengths ρ=0,0.5 and 1. Using these slowly varying L- and H-event amplitudes, we quantified the receptive field size for different levels of correlation strengths ρ between L- and H-event amplitudes (Figure 5—figure supplement 2A). We also quantified the proportion of simulations that resulted in selective, non-selective and decoupled receptive fields (Figure 5—figure supplement 2A, compare to Figure 3C in the manuscript). For the range of tested correlation strengths, less than 20% of simulations resulted in selective receptive fields, similar to the case for non-adaptive H-events (Figure 5—figure supplement 2A, compare to Figure 3C left in the manuscript).

2) We also considered whether a top-down signal that decays on a slow developmental timescale that affects only H-event amplitudes (without directly integrating ongoing activity in the network) might generate selectivity and prevent cortical decoupling. In these simulations, H-event amplitudes decay exponentially as a function of developmental time with different time constants (Figure 5—figure supplement 2B). We quantified the receptive field size for a range of decay time constants (Figure 5—figure supplement 2B). We also quantified the proportion of simulations that resulted in selective, non-selective and decoupled receptive fields (Figure 5—figure supplement 2B, compare to Figure 3C in the manuscript). Only very fast decay time constants resulted in selective and refined receptive fields. This scenario corresponds to H-events being effective only at the beginning of the simulation, which reduces to the case of pure L-events in our manuscript (gray box in Figure 5—figure supplement 2B). We do not consider this a biologically plausible scenario since H-events are detected in animals with ages differing by several days (P8 to P10) (Siegel et al., 2012).

This analysis is now included in the manuscript (Figure 5—figure supplement 3) and discussed in the second paragraph of the Results section “in vivo spontaneous cortical activity shows a signature of adaptation”.

Therefore, we conclude from our combination of data analysis, and the new simulations of receptive field refinements using alternative sources of slow activity fluctuations (either slowly varying correlated L and H amplitudes, or slowly decaying H-event amplitudes throughout development) that such mechanisms cannot generate the robust refinement of receptive fields that we find in our network model with the proposed adaptation of H-event amplitudes.